# In vivo imaging of the barrier properties of the glia limitans during health and neuroinflammation

Pauline Hélie-Legoupil[1,3,8], Florencia Kloster [1,8], Javier Pareja [1], Mykhailo Vladymyrov [1], Josephine A. Mapunda[1,4,5], Elisa Bouillet [1], Yannik Oetiker [1], Irene Spera[1,6], Sara Barcos[1], Amandine Brenna [1], Adolfo Odriozola[2], Alyssa Baert[1,7], Christoph Fankhauser[1], Beat Haenni[2], Steven T. Proulx [1], Benoît Zuber [2], Urban Deutsch[1] & Britta Engelhardt [1] ✉

The glia limitans ensheathes the entire central nervous system (CNS) parenchyma towards the outer surfaces and the perivascular spaces and is formed by a subset of astrocytes strategically localized at these outer parenchymal borders. Barrier properties of the glia limitans during health and neuroinflammation are incompletely understood. By developing an aquaporin-4 (Aqp4)-mRuby3 knock-in reporter mouse that allows for in vivo imaging of the superficial and perivascular glia limitans, we here show that the glia limitans forms a barrier for soluble mediators, beads and immune cells. Combining the Aqp4-mRuby3 reporter strain with additional reporter alleles for vascular, leptomeningeal or myeloid cells ensures precise localization of immune cells to CNS border zones versus the CNS parenchyma allowing to assign functional roles in CNS immune surveillance versus neuropathology. Availability of the Aqp4-mRuby3 reporter mouse will further advance our understanding of the active role of the glia limitans in CNS immune privilege.

The neurons in the central nervous system (CNS) coordinate all our body functions and rely on a homeostatic environment that does not tolerate uncontrolled entry of blood components. The CNS has therefore developed a unique relationship with the immune system known as CNS immune privilege (summarized in ref. 1). It was originally thought that antigens were sequestered in the CNS and thus invisible to the innate and adaptive immune systems and that CNS immune privilege was based on "immune ignorance". This concept was, however, abandoned, when in the 1980s it was shown that CSF drains into cervical lymph nodes (summarized in ref. 2) and that activated T cells can enter the CNS in the absence of neuroinflammation (summarized in refs. 3,4).

Revisiting CNS immune privilege, we have therefore proposed that the brain barriers have evolved for the protection and defense of the CNS parenchyma with an architecture that resembles that of a medieval castle surrounded by a two-walled castle moat[1,5]. In this concept, the outer brain barriers allow for controlled immune cell entry into CNS border zones (the moat), enabling CNS immune surveillance, while the inner brain barrier prohibits immune cell entry into the healthy CNS parenchyma, ensuring homeostasis for proper neuronal function[6]. This analogy is compatible with the known localization of the brain barriers: the endothelial blood-brain barrier (BBB) at the level of CNS parenchymal vessels, the epithelial blood-

[1]Theodor Kocher Institute, University of Bern, Bern, Switzerland. [2]Institute of Anatomy, University of Bern, Bern, Switzerland. [3]Present address: Division of Veterinary Anatomy, Vetsuisse Faculty, University of Bern, Bern, Switzerland. [4]Present address: School of Life Sciences and Bioengineering, Nelson Mandela African Institution of Science and Technology, Arusha, Tanzania. [5]Present address: The Hormel Institute, University of Minnesota, Austin, MN, USA. [6]Present address: Department of Biomedicine, University of Basel, Basel, Switzerland. [7]Present address: Institute of Neuroimmunology and Multiple Sclerosis Research, University Medical Center Göttingen, Göttingen, Germany. [8]These authors contributed equally: Pauline Hélie-Legoupil, Florencia Kloster. ✉e-mail: britta.engelhardt@unibe.ch

cerebrospinal fluid barrier (BCSFB) of the choroid plexuses (ChP) in the brain ventricles, the arachnoid barrier (AB) established by meningeal fibroblasts on the CNS surface and the glia limitans formed by astrocytes, which encloses the entire CNS parenchyma. In the castle concept, the CNS parenchyma is protected by the outer tight junction-fortified brain barriers (BBB, BCSFB, and AB), the cerebrospinal fluid (CSF)-filled castle moat (subarachnoid and perivascular space) that harbors border associated macrophages (BAMs) as "castle moat guards" and the inner wall constituted by the pia mater and glia limitans. Although formed by different cellular subtypes, namely fibroblasts, epithelial, and endothelial cells, the barrier properties of the AB, BCSFB, and BBB rely on a common cellular and molecular makeup. They are connected by unique bicellular and tricellular tight junctional complexes that prohibit uncontrolled paracellular diffusion of water-soluble molecules between the periphery and the CNS and thus establish a physical barrier[7–11]. In addition, the outer brain barriers constitute a metabolic barrier due to the expression of unique combinations of enzymes, transporters and efflux pumps that enable the high metabolic demands of the neuronal cells in the CNS parenchyma to be met and to ensure rapid efflux of toxic metabolites from the CNS[10–12].

In contrast, the glia limitans, which is in evolution the oldest brain barrier in vertebrates[13], does not entirely prohibit passage of soluble molecules but does provide a barrier for immune cells[14], (summarized in ref. [1,4]). Thus, during evolution, it has turned into a second brain barrier "behind" the outer brain barriers in mammals. The glia limitans is formed by a subset of astrocytes strategically positioned at the superficial and perivascular borders of the brain and spinal cord parenchyma. The perivascular glia limitans is characterized by polarized astrocytes that extend their endfeet towards the perivascular space and deposit a parenchymal basement membrane, which is molecularly distinct from the endothelial basement membrane[15]. The superficial glia limitans is, however, rather formed by a layer of evolutionarily conserved myocilin-expressing surface astrocytes displaying a unique morphology[16,17]. The basement membrane of the superficial glia limitans is shared by the astrocytes and pial fibroblasts[10].

While the role of the outer brain barriers in controlling immune cell entry into the CNS has been addressed, the role of the glia limitans-forming astrocytes in neuroimmune communication maintenance is not well explored. Studies in experimental autoimmune encephalomyelitis (EAE), an animal model of multiple sclerosis, have provided evidence that clinical disease only starts after immune cells have crossed the glia limitans and reached the CNS parenchyma proper[6]. This assigns an important role for the glia limitans in maintaining CNS immune privilege and homeostasis.

Intravital microscopy of immune cell interactions with the brain barriers has significantly advanced our understanding of CNS immunity. To explore the role of the glia limitans in CNS immune surveillance and neuroinflammation we have therefore developed a fluorescent reporter mouse allowing for in vivo imaging of the glia limitans. To this end, we made use of the observation that expression of the water channel aquaporin-4 (AQP4) is strongly polarized at the endfeet of perivascular astrocytes[18] and highly expressed in astrocytes forming the superficial glia limitans[19]. We have therefore developed an Aqp4-mRuby3 knock-in reporter mouse that is suitable for in vivo visualization of the perivascular and superficial glia limitans of the brain and spinal cord in mice. We here show that the Aqp4-mRuby3 knock-in reporter mouse allows for exploring the role of the glia limitans in controlling immune mediator distribution and immune cell trafficking within the CNS and provides thus a valuable tool to advance our understanding of the role of the glia limitans in neuroimmune interactions in the CNS during health and disease.

## Results

### The Aqp4-mRuby3 knock-in mouse allows for ex vivo visualization of the perivascular and superficial glia limitans

To allow for in vivo visualization of the glia limitans we took advantage of the polarized expression of the water channel aquaporin 4 (AQP4) in perivascular and subpial astrocytes forming the glia limitans. Based on the successful design of an Aqp4-GFP fusion protein described previously[20], we introduced the open reading frame of the red fluorescent protein mRuby3 into exon 2 of the aquaporin 4 (Aqp4) gene coding for the second extracellular loop leading to expression of an AQP4-mRuby3 fusion protein in these Aqp4-mRuby3 knock-in reporter mice (Fig. 1a, b; Supplementary Fig. 1). This AQP4-mRuby3 fusion protein should be exposed on the cell surface of astrocytes and participate in the formation of higher order complexes such as orthogonal arrays of particles (OAPs) as described earlier[20]. Aqp4-mRuby3 knock-in mice did not show any differences in fertility and litter size when compared to their wild-type (WT) counterparts and were born at mendelian ratios. Expression and correct localization of the AQP4-mRuby3 fusion protein in perivascular astrocyte endfeet and in astrocytes of the superficial glia limitans was confirmed by the colocalization of immunostaining for AQP4 in brain and spinal cord in cryosections of decalcified heads and vertebral columns from heterozygous Aqp4-mRuby3 mice (Fig. 1c, d). In addition, similar expression patterns of AQP4 protein were observed in WT C57BL/6 J mice, confirming the correct expression and localization of the AQP4-mRuby3 fusion protein within the CNS of Aqp4-mRuby3 reporter mice (Supplementary Fig. 2a, b). Immunostainings for laminin, podocalyxin and GFAP verified the localization of the endogenous AQP4-mRuby3 signal adjacent to parenchymal basement membranes, vascular endothelial cells and astrocytes, respectively (Supplementary Fig 2c–f). Finally, we also confirmed the expression of the reporter Aqp4-mRuby3 at the glia limitans from the inner blood-retina barrier (Supplementary Fig. 2g). Thus, the AQP4-mRuby fusion protein correctly localized to astrocytes forming the perivascular and superficial glia limitans.

### Brain and spinal cord water balance during health and neuroinflammation is not affected in heterozygous Aqp4-mRuby3 knock-in reporter mice

Since functional AQP4 forms tetramers when assembled in the plasma membrane and higher order OAPs[20], we asked if expression of the AQP4-mRuby3 fusion protein might affect assembly and function of the AQP4 water channel and therefore compromise CNS water balance. To this end, we compared the water content of the brain and spinal cord of both homozygous and heterozygous Aqp4-mRuby3 reporter mice with their WT counterparts. We observed a small but significant increase in brain water content of homozygous Aqp4-mRuby3 mice when compared to heterozygous Aqp4-mRuby3 and WT littermates (Supplementary Fig. 3). At the same time, we did not detect any significant differences in brain water content of heterozygous Aqp4-mRuby3 mice when compared to WT littermates. Also, spinal cord water content was not affected by expression of the AQP4-mRuby3 fusion protein (Supplementary Fig. 3). These data suggest that the absence of WT AQP4 in homozygous Aqp4-mRuby3 knock-in reporter mice may affect brain water balance in healthy mice and compelled us to use heterozygous mice for all further analysis.

As involvement of AQP4 in neuroinflammation has been described[21], we explored if heterozygous Aqp4-mRuby3 mice would display an altered pathogenesis of experimental autoimmune encephalomyelitis (EAE), an animal model of multiple sclerosis (MS) compared to WT littermates (Fig. 2a). We found no difference in overall clinical severity of EAE between female heterozygous Aqp4-mRuby3 reporter mice and WT controls within 30 days after EAE induction (Fig. 2b). Furthermore, we verified that expression of Aqp4-mRuby3 in female heterozygous mice did not affect the water

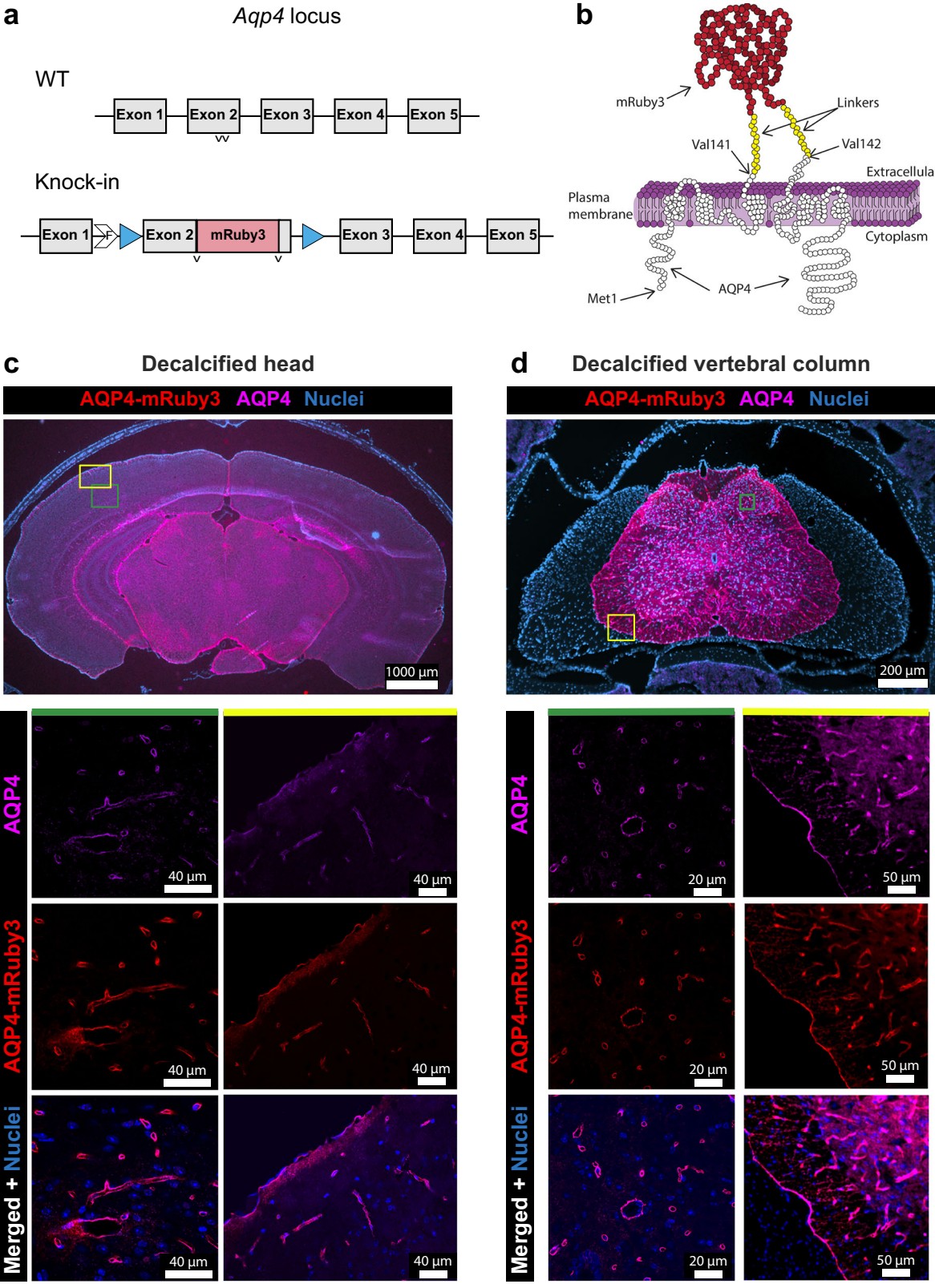

content of the brain and spinal cord compared to WT mice during the course of EAE (Fig. 2c, d). Also, splenocytes isolated from female heterozygous Aqp4-mRuby3 and WT mice at day 30 after EAE induction showed no differences in their proliferation upon recall by increasing concentrations of the encephalitogenic MOG$_{35-55}$ peptide (Fig. 2e). Thus, expression of the AQP4-mRuby3 fusion protein in heterozygous mice does not affect EAE pathogenesis.

Loss of polarized localization of AQP4 at perivascular astrocyte endfeet has been observed in many neurological disorders, including EAE[22]. Exploring the localization of the AQP4-mRuby3 signal in brain sections of female mice with EAE we here confirmed a comparable loss of the polarized AQP4-mRuby3 signal from perivascular astrocyte endfeet around inflamed brain microvessels (Fig. 2f). We furthermore confirmed co-localization of the endogenous mRuby3 signal with an

**Fig. 1 | Characterization of the Aqp4-mRuby3 knock-in reporter mouse.**
**a** Construct of the Aqp4-mRuby3 knock-in mouse: a monomeric mRuby3 fluorescent protein was inserted between valines 141 and 142 (v) in the exon 2 of the endogenous *Aqp4* gene. The mRuby3 sequence was flanked by flexible GlySer-rich linkers to exclude protein folding constraints. F denotes the remaining FRT site after removal of the neo cassette. The full knocked-in exon 2 was additionally flanked by loxP and lox2272 sites (blue triangles). **b** Schematic representation of the AQP4-mRuby3 fusion protein. The mRuby3 (red) was flanked by two GlySer-rich linkers (yellow) and inserted between valines 141 and 142 of the AQP4 protein (white). **c, d** Images of 20 μm coronal cryosections of a decalcified head (**c**) and lumbar vertebral column (**d**) of an Aqp4-mRuby3 mouse. Top panels show

overviews acquired by epifluorescence microscopy. **c** The AQP4-mRuby3 signal from the Aqp4 knock-in allele is shown in red, antibody-mediated immunostaining for the AQP4 protein in purple, and nuclei stained with DAPI are seen in blue. Areas boxed in green (brain cortex) and yellow (brain meninges) show confocal images in higher magnification from (**c**). **d** The AQP4-mRuby3 signal from the Aqp4 knock-in allele is shown in red, antibody-mediated immunostaining for the AQP4 protein in purple, and nuclei stained with DAPI are seen in blue. Areas boxed in green (spinal cord parenchyma) show confocal images in higher magnification from (**d**). Areas boxed in yellow (spinal cord meninges) show epifluorescence images in higher magnification from (**d**). Data are representative of 3 mice.

anti-AQP4 immunostaining signal around the brain parenchymal blood vessels during EAE (Fig. 2f). The inflammatory status of the brain parenchymal blood vessels associated with loss of polarized AQP4-mRuby localization was characterized by the presence of infiltrating leptomeningeal and perivascular CD45$^+$ immune cells (Fig. 2g). Quantification of the perivascular cuffs in the brains of mice during EAE confirmed a comparable number between WT and Aqp4-mRuby3 mice (Fig. 2h).

Taken together, these data show that heterozygous expression of AQP4-mRuby3 does neither affect brain and spinal cord water balance during health and neuroinflammation nor EAE pathogenesis.

## Heterozygous Aqp4-mRuby3 reporter mice display intact ultrastructural astrocyte foot morphology

Perivascular astrocytes contribute to blood-brain barrier integrity[23]. Therefore, we next explored if heterozygous expression of the AQP4-mRuby3 fusion protein affected astrocyte end-foot and BBB morphology at the ultrastructural level. To this end, we performed transmission electron microscopy of ultrathin brain sections from heterozygous Aqp4-mRuby3 and WT mice and investigated astrocyte end-foot morphology in the neocortex, the cerebellum and the brainstem. Blinded visual analysis allowed for classification of 3 types of morphologies of perivascular astrocyte endfeet which we categorized as normal (swelling score = 0), slightly enlarged/abnormal (swelling score = 1) and enlarged/abnormal (swelling score = 2), based on the size of the astrocyte end-foot and the appearance of the cytoplasm (Fig. 3a). Quantification of the respective end-foot morphologies did not reveal any significant differences between WT and heterozygous Aqp4-mRuby3 reporter mice when comparing the swelling scores across different brain areas (Fig. 3b). The swelling score distributions were comparable across individual mice within each genotype, which allowed us to exclude potential sample preparation artifacts that could differentially affect the astrocyte end-foot morphology and mask an effect in heterozygous Aqp4-mRuby3 mice (Fig. 3c). Overall, we concluded that heterozygous expression of AQP4-mRuby3 does not significantly affect astrocyte end-foot and CNS vascular morphology at the ultrastructural level.

## Heterozygous Aqp4-mRuby3 reporter mice allow for in vivo imaging of the glia limitans

We next explored the suitability of the heterozygous Aqp4-mRuby3 reporter mice for in vivo detection of the glia limitans in the brain and spinal cord by performing 2-photon intravital microscopy (2P-IVM) through cranial window or skull thinning preparations placed over the cortical surface of the right brain hemisphere or through a cervical spinal cord window (Fig. 4a, c, e). The vascular lumen was visualized by injection of a 10 kDa FITC-dextran during 2P-IVM. By using an excitation wavelength of $\lambda = 1045$ nm, the AQP4-mRuby3 signal was detected at the expected localization of the perivascular and superficial glia limitans in the brain (Fig. 4b, d, Supplementary Movies 1, 2). Importantly, 2P-IVM imaging with this excitation wavelength will create the second harmonic generation (SHG) signal from the type I collagen in the dura mater and the remaining skull bone at an emission

wavelength of $\lambda = 523$ nm and thus in the spectrum of green visible light. In fact, the perivascular glia limitans of the brain was outlined by a very bright AQP4-mRuby3 signal according to the highly polarized expression of AQP4 at astrocyte endfeet, while the superficial glia limitans was visible with a more diffuse AQP4-mRuby3 signal from the superficial astrocytes (Fig. 4b, d).

To verify if the difference in the pattern of the observed AQP4-mRuby3 signal at the perivascular and superficial glia limitans during 2P-IVM was due to the reported difference of astrocyte morphology, we performed serial block-face scanning electron microscopy (SBF-SEM) of the brains from heterozygous Aqp4-mRuby3 and WT mice and investigated astrocyte morphology at the superficial glia limitans. In accordance to previous reports[16,17] we found astrocyte cell bodies interspersed with astrocyte processes rather than astrocyte endfeet forming the superficial glia limitans (Supplementary Fig. 4 and Supplementary Movie 3).

Having verified the different astrocyte morphology underlying the superficial versus the perivascular glia limitans at the ultrastructural level, we next asked whether the localization of AQP4-mRuby3 is polarized towards the outer surface on the superficial astrocytes. Confocal imaging showed that AQP4-mRuby3 is polarized at the outer surface of GFAP$^+$ superficial astrocytes, rather than distributed uniformly throughout their cell bodies (Supplementary Fig. 5a, b). In the transition between the superficial and perivascular glia limitans, we detected GFAP$^+$ astrocyte cell bodies surrounded by GFAP$^+$ processes, potentially the first astrocyte endfeet forming the perivascular glia limitans. Importantly, confocal imaging showed a continuous AQP4-mRuby signal from the superficial to the perivascular glia limitans polarized towards the outer surface of the CNS parenchyma (Supplementary Fig. 5a, b).

We also detected by 2P-IVM the AQP4-mRuby3 signal on the surface of the cervical spinal cord parenchyma and following the ascending venules, thus identifying the spinal cord superficial and perivascular glia limitans, respectively (Fig. 4e, f, Supplementary Movie 4). SHG highlighted the spinal cord dura mater, the trabeculae of the spinal cord subarachnoid space (SAS) and subpial collagen. Interestingly, the AQP4-mRuby3 signal appeared discontinuous underneath the dorsal spinal cord vein. Therefore, we performed epifluorescence imaging of spinal cord cryosections from Aqp4-mRuby3 mice, which showed a continuous localization of the AQP4-mRuby3 signal in the dorsal aspect of the spinal cord, including the area under the dorsal vein (Supplementary Fig. 6). These data show that the apparent discontinuity in the AQP4-mRuby3 signal observed with 2P-IVM is a consequence of the light absorption and optical properties of the blood within the dorsal vein. Taken together, our data underline the suitability of the heterozygous Aqp4-mRuby3 mice for in vivo imaging of the superficial and perivascular glia limitans of the brain and spinal cord.

## Establishing a CNS border reporter mouse

We have previously identified a reporter mouse, the VE-cadherin-GFP knock-in line, as a novel tool for in vivo imaging of the leptomeningeal layers in addition to endothelial cell junctions[24]. Aiming to create a CNS

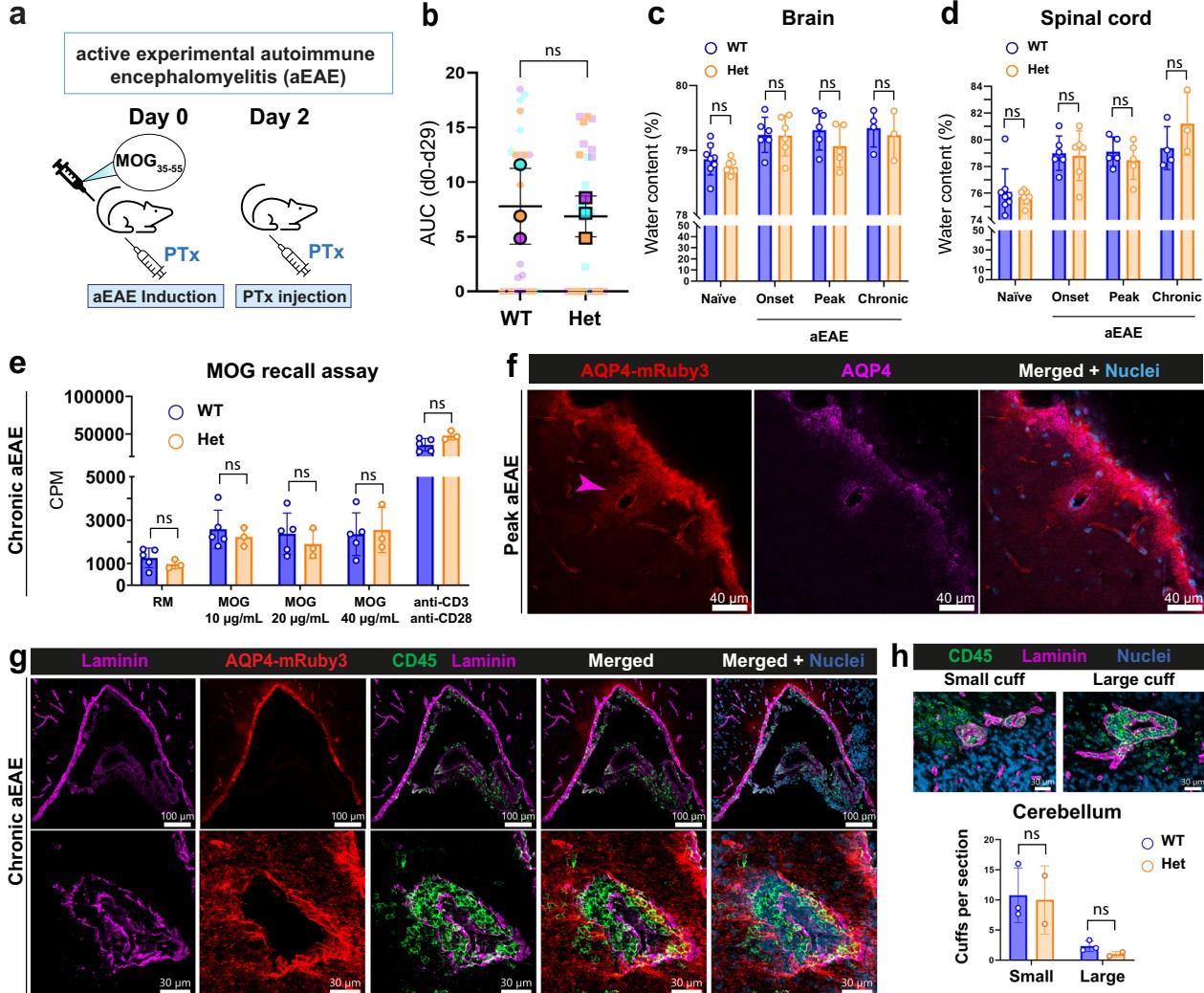

**Fig. 2 | AQP4-mRuby3 expression in heterozygous mice does not affect neuroinflammation. a** Scheme of aEAE experiment. MOG = myelin oligodendrocyte glycoprotein; PTx = pertussis toxin. **b** Overall aEAE disease severity from day of immunization (day 0) to day 29 determined by the area under the curve (AUC) of female WT and heterozygous (Het) Aqp4-mRuby3 reporter mice (3 independent experiments, n(WT) = 27 and n(Het)=28). SuperPlots[83] are shown, where each point represents an individual mouse and outlined points represent the mean of each experiment. Data shown as mean ± SD and analyzed using a two-tailed paired t-test. **c, d** Water content (in %) after drying process of 24 h in brains (**c**) and spinal cords (**d**) of female heterozygous Aqp4-mRuby3 and WT control mice at different clinical stages of aEAE, from 3 independent experiments. n(WT Naïve)=8, n(Het Naïve)=7; n(WT onset)=6, n(Het onset)=6; n(WT peak)=5, n(Het peak)=5; n(WT chronic)=4, n(Het chronic)=3. Data shown as mean ± SD and analyzed using two-way ANOVA with Šidák's multiple comparisons test. **e** Splenocytes from female WT (n = 5) and Het Aqp4-mRuby3 mice (n = 3) were isolated at day 30 of aEAE. Incorporation of 3H-thymidine as a measure for cell proliferation in counts per minute (CPM) of splenocytes in medium (RM) as a negative control, in the presence of 10, 20, and 40μg/mL of MOG35-55, or polyclonal anti-CD3 and CD28 antibody induced polyclonal proliferation as a positive control. Data shown as mean ± SD and analyzed by unpaired parametric t-tests. Source data are provided as a Source Data file.
**f, g** Immunofluorescence staining of 20 μm thick cryosections from female Aqp4-mRuby3 mice during aEAE (n = 2). **f** Staining for AQP4 in mouse skull cryosection at peak of aEAE (score 1). Pink arrowhead shows the loss of the perivascular polarized AQP4-mRuby signal during neuroinflammation. **g, h** Laminin (magenta) and CD45 (green) immunofluorescence stainings of brain cryosection of mouse at chronic stage of aEAE (score 0.5 to 1). **h** Quantification of perivascular cuffs in WT (n = 3) and Aqp4-mRuby3 mice (n = 2) from 1 aEAE experiment. Top panel shows representative images of small and large perivascular cuffs observed in immunofluorescence stainings from (**g**). Data shown as mean ± SD, two-way ANOVA with Šídák multiple comparisons test was performed.

border reporter mouse that will allow for simultaneous imaging of the vascular, pial and arachnoid CNS borders (VE-cadherin-GFP) and the glia limitans (Aqp4-mRuby3) we crossed the Aqp4-mRuby3 mice with VE-cadherin-GFP knock-in reporter mice. Confocal imaging of decalcified head cryosections from the CNS border reporter mice (Aqp-mRuby3; VE-cadherin-GFP mice) showed the AQP4-mRuby3 signal from the superficial glia limitans to be localized right below the leptomeningeal VE-cadherin-GFP signal (Fig. 5a) and the perivascular AQP4-mRuby3 signal bordering the perivascular space around VE-cadherin-GFP expressing vascular walls (Fig. 5b). To explore simultaneous in vivo imaging of the glia limitans and the leptomeningeal and endothelial borders we developed 2P-IVM modalities of the brain and spinal cord in CNS border reporter mice through cranial window, skull thinning and spinal cord window preparations (Fig. 6, Supplementary Movie 5 (cranial window), Supplementary Movie 6 (spinal cord window)). Efficient 2P excitation of Aqp4-mRuby3 required a wavelength range of λ1 = 950-1100 nm, with a peak at λ1 = 1045 nm. At the same time, 2P excitation peak for VE-cadherin-GFP was reached at λ2 = 920 nm, with its signal gradually decreasing with increasing excitation wavelengths. Therefore, simultaneous in vivo imaging of the AQP4-mRuby3 and VE-cadherin-GFP signals in the CNS border reporter mice required the adoption of a novel imaging setup using two excitation wavelengths, namely of λ1 = 1045 nm and λ2 = 920 nm. Notably, this imaging setup generates a SHG signal in both the blue and green

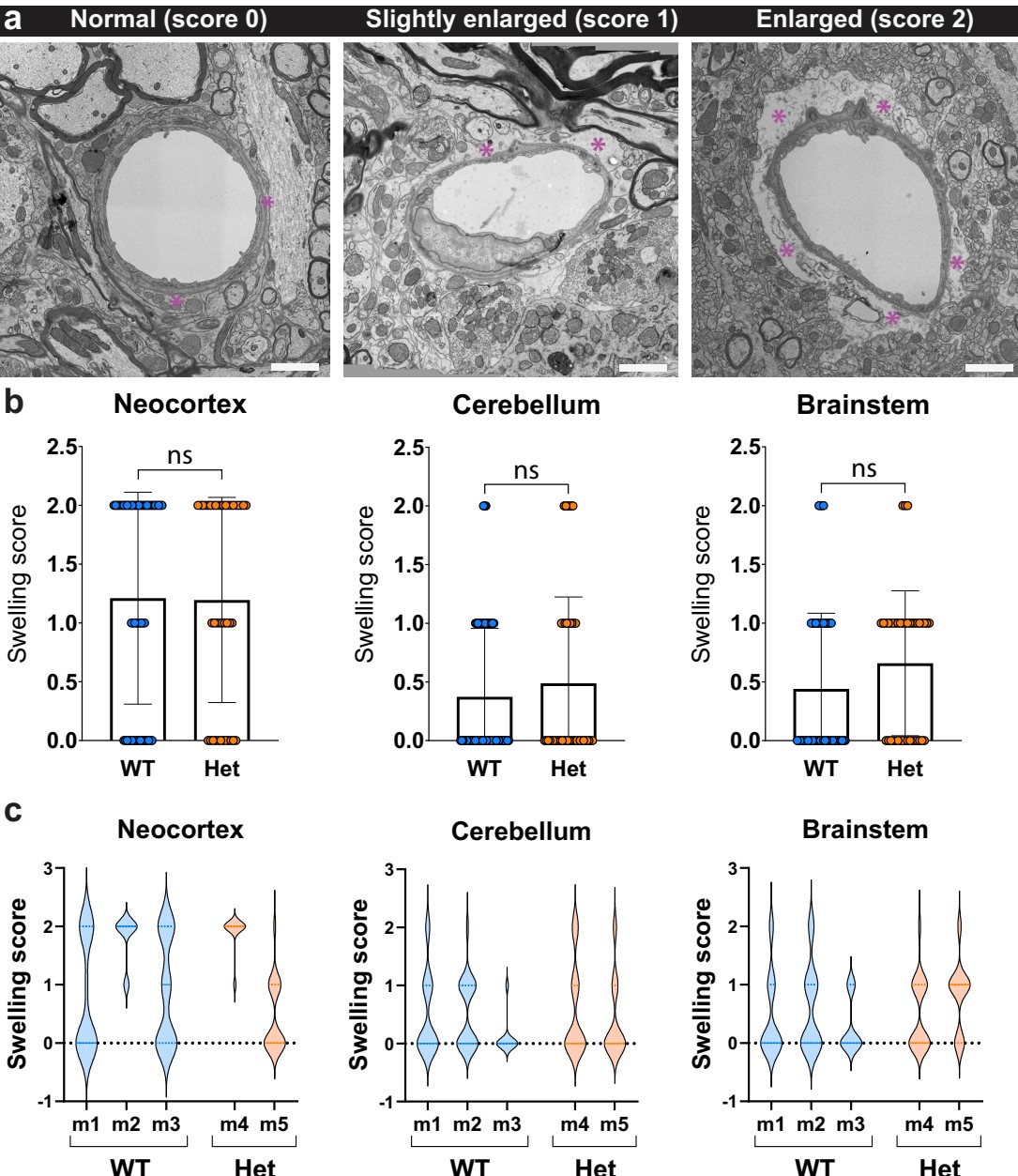

**Fig. 3 | Heterozygous Aqp4-mRuby3 reporter mice display intact ultrastructural astrocyte end-foot morphology. a** Representative images from normal appearing (score 0), slightly enlarged (score 1) and enlarged (score 2) astrocyte endfeet based on their analysis at the ultrastructural level by transmission electron microscopy (TEM). Assignment of astrocyte endfeet to the respective categories was performed in a blinded fashion as outlined in Methods. Asterisks indicate astrocytic endfeet. Scale bar: 2 μm. **b** Swelling score from astrocytic endfeet in neocortex, cerebellum, and brainstem of WT (blue) and heterozygous Aqp4-mRuby3 mice (orange) is shown. Data are shown as mean ± SD. Data were analyzed using a two-sided parametrical t-test. Data are representative of 57, 59, and 50 vessels from the neocortex, cerebellum, and brainstem, respectively, from 3 WT mice and 41, 43, and 41 vessels from the neocortex, cerebellum, and brainstem, respectively, from 2 heterozygous mice (2 independent experiments). **c** Violin plots display astrocyte endfeet swelling score distribution from (**b**) for each individual mouse of both genotypes in blood vessels from the neocortex, cerebellum, and brainstem. Source data are provided as a Source Data file.

light spectrum, corresponding to the excitation with λ2 = 920 nm and λ1 = 1045 nm, respectively.

This imaging setup allowed for the detection of the AQP4-mRuby3 signal and visualization of the superficial and perivascular glia limitans of the brain (Fig. 6a, b) and spinal cord (Fig. 6c) in the CNS border reporter mice. Importantly, the AQP4-mRuby3 signal showed funnel-like patterns at the level of VE-cadherin-GFP+ blood vessels diving into or emerging from the CNS parenchyma, where the glia limitans transitions from the superficial to the perivascular glia limitans thus enabling precise definition of the outer border of the perivascular spaces (Fig. 6, Supplementary Fig. 5c). Importantly, 2P-IVM confirmed a continuous AQP4-mRuby signal at the transition zone from the superficial to the perivascular glia limitans polarized towards the outer surface of the CNS parenchyma in vivo (Supplementary Fig. 5c). In addition, the AQP4-mRuby3 signal allowed for precise localization of the superficial glia limitans just below the layer of VE-cadherin-GFP+ pia mater fibroblasts (Fig. 6). It is important to note that imaging with excitation wavelengths of λ1 = 1000-1100 nm results in SHG in the green light spectrum, thus limiting distinction of SHG as landmark for skull bone and dura mater from the VE-cadherin-GFP signal of blood vessels and meningeal fibroblasts. While this provides no limitation for simultaneous 2P-IVM of the CNS

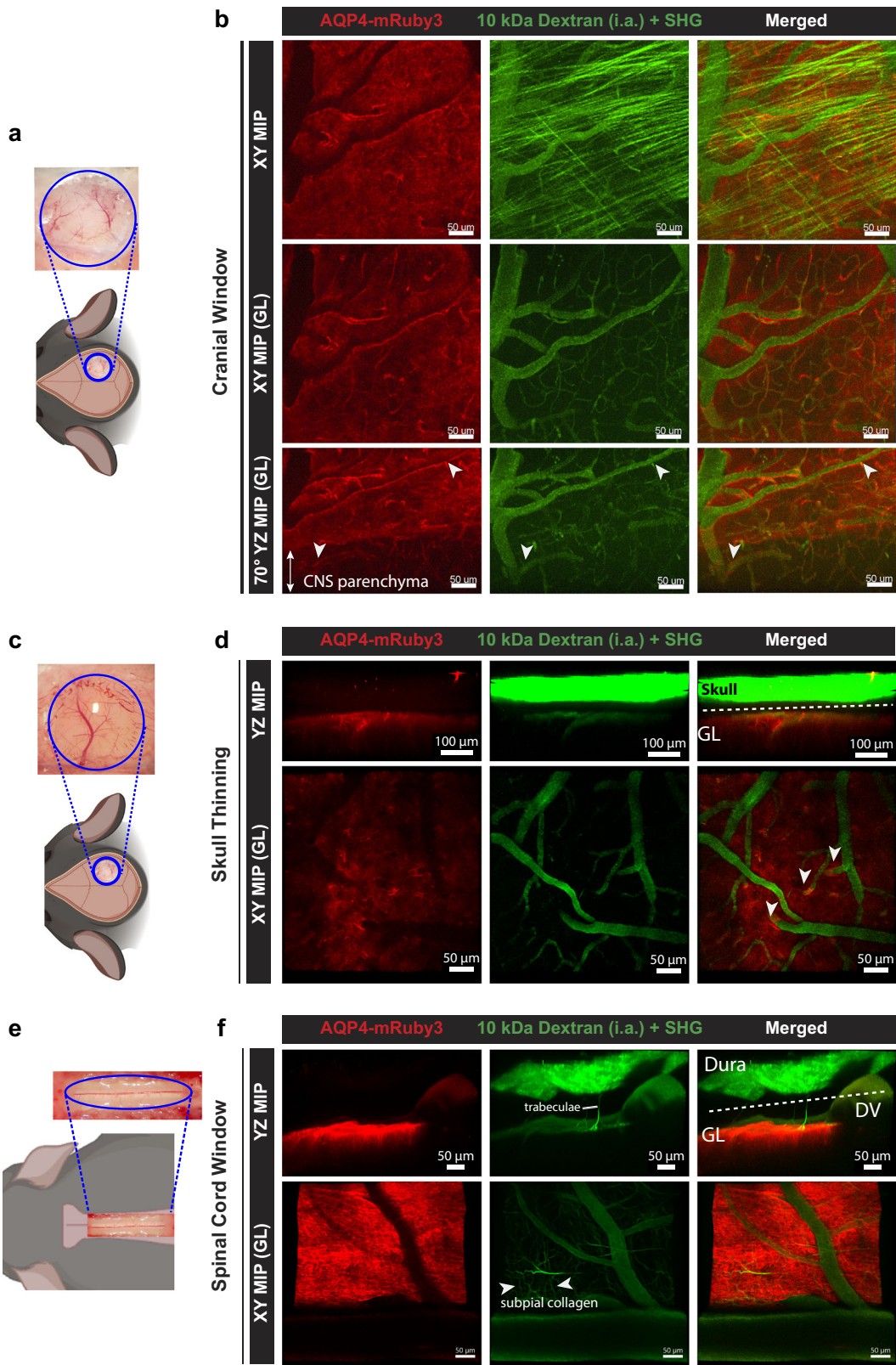

borders in the spinal cord, the intense SHG signal from the bone and dura mater obtained when imaging the brain through a thinned skull prohibits the unequivocal identification of the localization of the VE-cadherin-GFP expressing meningeal fibroblasts (Fig. 6b). Taken together, the CNS border reporter mouse developed by crossing Aqp4-mRuby3 and VE-cadherin-GFP reporter mice allows for simultaneous imaging of the glia limitans and the vascular and leptomeningeal borders and thus for direct visualization of the CNS zonation established for CNS immune surveillance.

## AQP4-mRuby3 marked glia limitans allows classification of CNS resident myeloid cell subsets

The glia limitans contributes to the zonation of the CNS that establishes compartments, which harbor distinct subsets of CNS resident

**Fig. 4 | The Aqp4-mRuby3 knock-in reporter mouse allows for in vivo imaging of the glia limitans. a, c, e** Schemes showing localization and the respective surgical preparations of the cranial window (**a**), skull thinning (**c**) and spinal cord window (**e**) for 2P-IVM imaging of the Aqp4-mRuby3 knock-in reporter mouse. 2P excitation wavelengths of 920 nm and 1045 nm were used simultaneously. Created in BioRender. Kloster, F. (2025) https://BioRender.com/dh3qfvt. **b** Representative images of 2P-IVM of the brain surface through a cranial window of Aqp4-mRuby3 mice after infusion of 10 kDa-FITC dextran (green) via a carotid artery catheter. Second harmonic generation (SHG) is visible in green at this excitation wavelength, superposing the collagen type I fibers (green) over the FITC dextran-filled vascular lumens (green). XY maximum intensity projections (MIP) from the whole Z-stack

(top panel) and from the cortical surface excluding the dura mater (middle panel) are shown. Bottom panel shows a 70° angle YZ MIP from the cortical surface, with the AQP4-mRuby3 signal coating the vessels that are diving into the parenchyma (white arrowheads). **d** Representative images of 2P-IVM of the brain surface of Aqp4-mRuby3 mice through a thinned skull. XY and YZ MIPs from the surface of the brain are shown. White arrows point to AQP4-mRuby3 signal following vessels within the brain cortex. **f** Representative images of 2P-IVM of the spinal cord surface of Aqp4-mRuby3 mice through a cervical spinal cord window. YZ and XY MIPs from the surface of the spinal cord are shown. Data are representative of 4 independent mice for each preparation. DV dorsal vein, GL glia limitans, MIP maximum intensity projection, i.a. intra-arterial, SHG second harmonic generation.

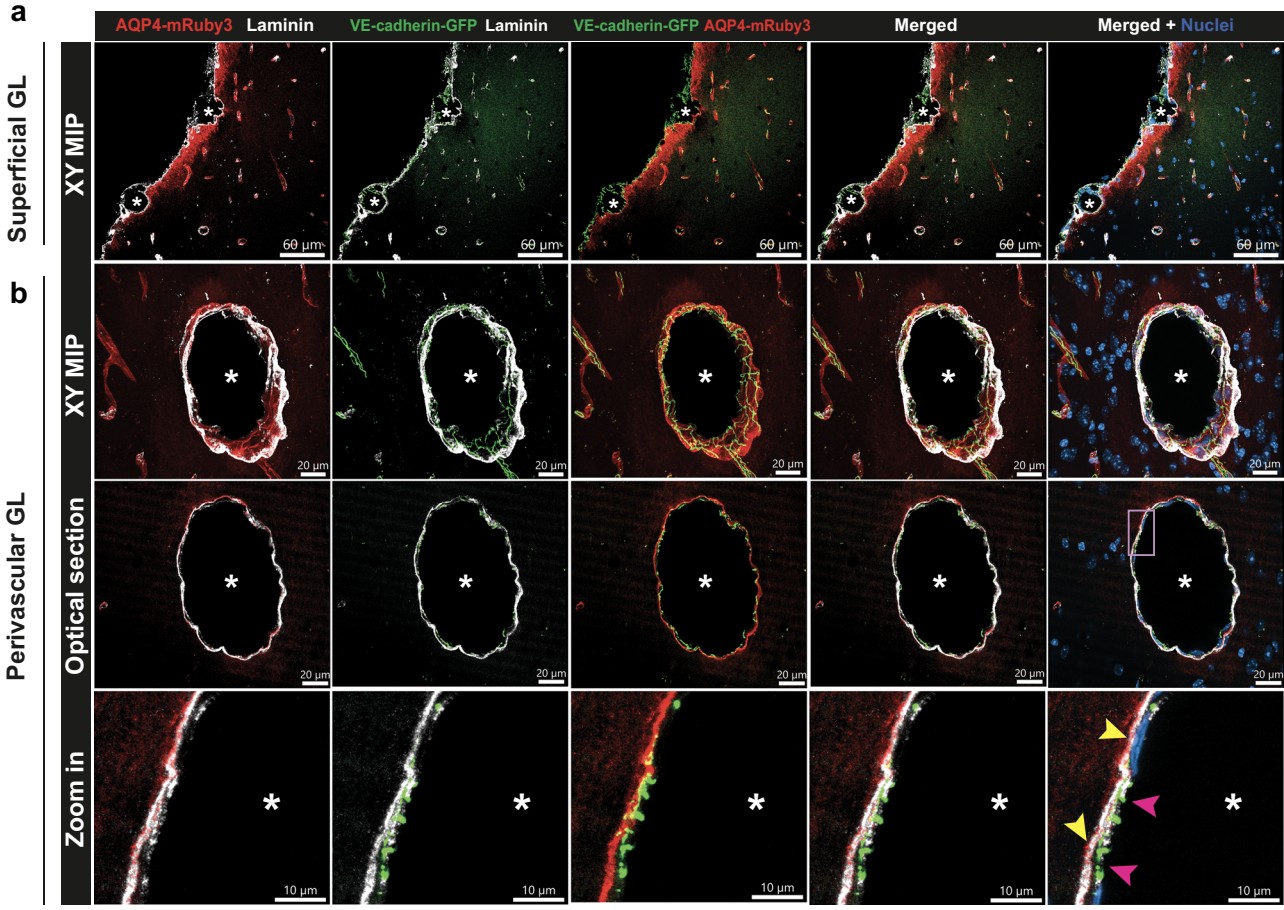

**Fig. 5 | Characterization of the CNS border reporter (Aqp4-mRuby3; VE-cadherin-GFP) mouse.** Confocal images of immunofluorescence stainings of 20 µm thick brain cryosections collected from healthy Aqp4-mRuby3; VE-cadherin-GFP CNS border reporter mice. The AQP4-mRuby3 signal is seen in red and the GFP⁺ adherens junctions of the blood vessels and between leptomeningeal fibroblasts are seen in green. **a, b** Immunofluorescence staining for laminin (white) and DAPI staining for nuclei. Asterisks mark the lumen of leptomeningeal (**a**) or parenchymal

blood vessels (**b**); n = 2. **a** XY MIP shows that the parenchymal basement membrane (white) colocalizes with the AQP4-mRuby3 signal from the superficial glia limitans (GL). VE-cadherin-GFP⁺ leptomeninges lie above of the GL. **b** First row, XY MIP of penetrating blood vessel. Second row, optical section of 2.2 µm. Third row, inset of ROI indicated by pink square. The VE-cadherin-GFP signal from endothelial cells (pink arrowheads) is surrounded by AQP4-mRuby3 signal from the perivascular GL (yellow arrowheads) and colocalizes with the parenchymal basement membrane.

myeloid cells[25]. Although it has become evident that border associated macrophages (BAMs) residing in the CSF-filled SAS and perivascular spaces, fulfill different functions compared to the parenchymal microglial cells, many in vivo imaging studies using CX3CR1-GFP myeloid reporter mice have failed to unequivocally distinguish CX3CR1-GFP expressing microglia from BAMs[26,27]. We here crossed the Aqp4-mRuby reporter mouse with CX3CR1-GFP myeloid reporter mice, which allowed us to image CX3CR1-GFP⁺ myeloid cells in spatial correlation to the glia limitans in the brain (Fig. 7a–e) and the spinal cord (Fig. 7f–j). Cranial window 2P-IVM allowed for detection of CX3CR1-GFP⁺ dural macrophages, BAMs in the SAS and perivascular spaces, and microglia within the CNS parenchyma below the AQP4-

mRuby3⁺ glia limitans (Fig. 7a–e) and occasional circulating CX3CR1-GFP expressing monocytes previously reported[28].

Making use of the AQP4-mRuby3 signal from the glia limitans and SHG from the collagen type I fibers in the dura mater allowed for assigning the myeloid cell subsets to their respective CNS compartments (Fig. 7b and Supplementary Movie 7). A group of morphologically distinct round-shaped CX3CR1-GFP⁺ macrophages were homogenously distributed at the expected level of dural collagen layers, as visualized by faint SHG in green due to the excitation wavelength of λ=1000 nm used in this setting (Fig. 7c). The brighter GFP signal obtained in the CX3CR1-GFP myeloid reporter mice compared to that in the VE-cadherin-GFP reporter mice allowed for

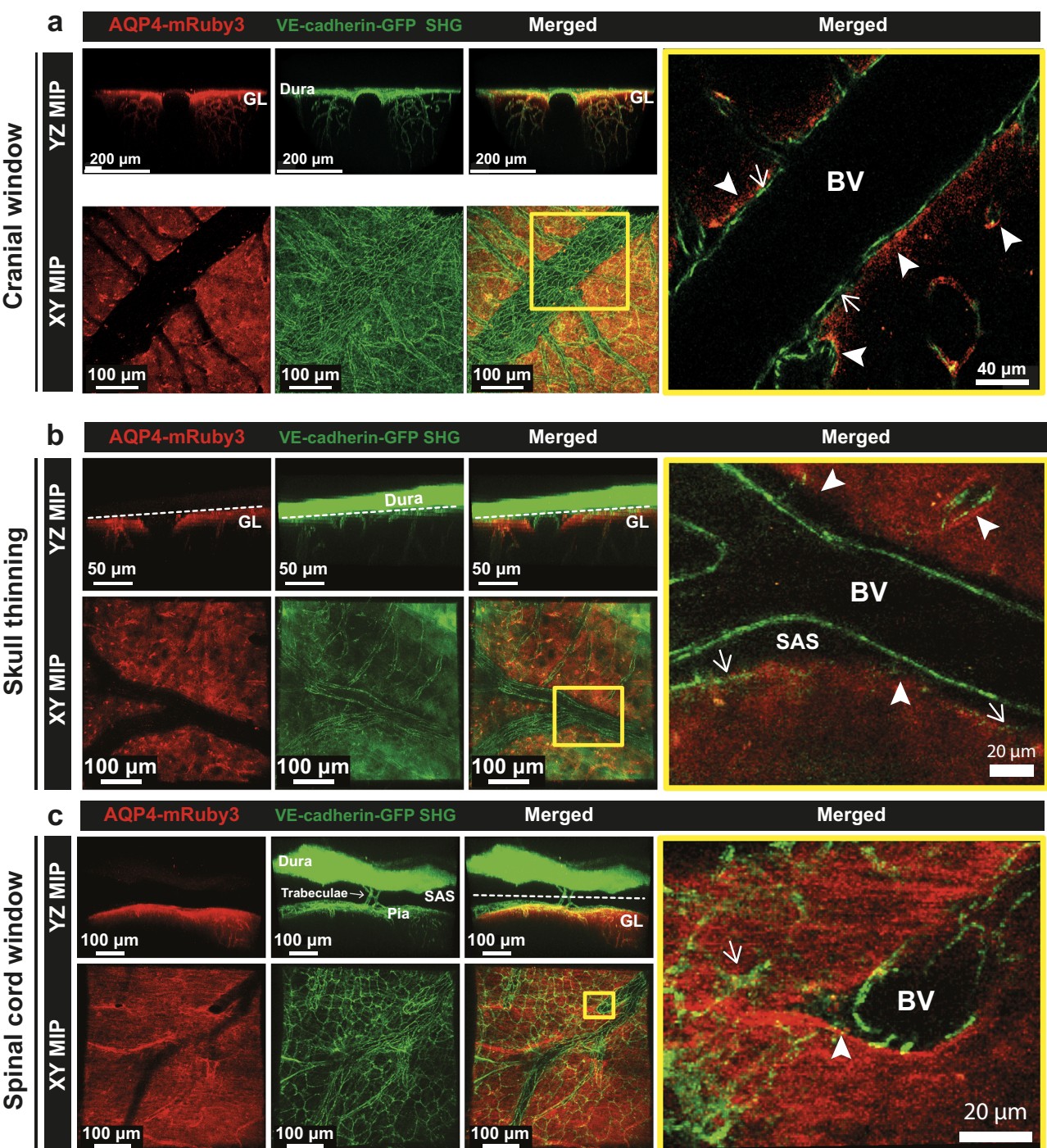

**Fig. 6 | Simultaneous in vivo imaging of the glia limitans and the vascular and leptomeningeal CNS borders.** Aqp4-mRuby3 mice were crossed with VE-cadherin-GFP mice creating CNS border reporter mice (Aqp4-mRuby3; VE-cadherin-GFP). Representative images of 2P-IVM imaging of the brain and spinal cord surface of CNS border reporter mice through cranial window (**a**), skull thinning (**b**) and cervical spinal cord window (**c**) are shown. 2P excitation wavelengths of 920 nm and 1045 nm were used simultaneously. **a** Representative YZ and XY MIP 2P images of the brain surface through a cranial window of a CNS border reporter mouse. Right panel shows a 2 μm thick optical section from the yellow inset on the merged XY MIP image. The AQP4-mRuby3 signal (red) allows for imaging of the glia limitans (white arrowheads) and the VE-cadherin-GFP signal (green) depicts the adherens junctions from the endothelial cells and the leptomeningeal fibroblasts (white arrows). **b, c** Representative YZ MIP 2P images of the brain (**b**) and spinal cord (**c**) surface through a skull thinning (**b**) and a spinal cord window (**c**) of a CNS border reporter mouse are shown in the top panels. Bottom panels show representative XY MIP 2P images excluding the skull and dura mater (**b**) or the dura mater (**c**) following the dotted line at the YZ MIP images. Right panel shows a 2 μm thick optical section from the yellow inset on the merged XY MIP image. The AQP4-mRuby3 (red) signal allows for imaging of the glia limitans (white arrowheads) and the VE-cadherin-GFP signal (green) depicts the adherens junctions from the endothelial cells and the leptomeningeal fibroblasts (white arrows). Data are representative of 3 mice per condition. GL glia limitans, BV blood vessel lumen, SAS subarachnoid space, MIP maximum intensity projection, SHG second harmonic generation.

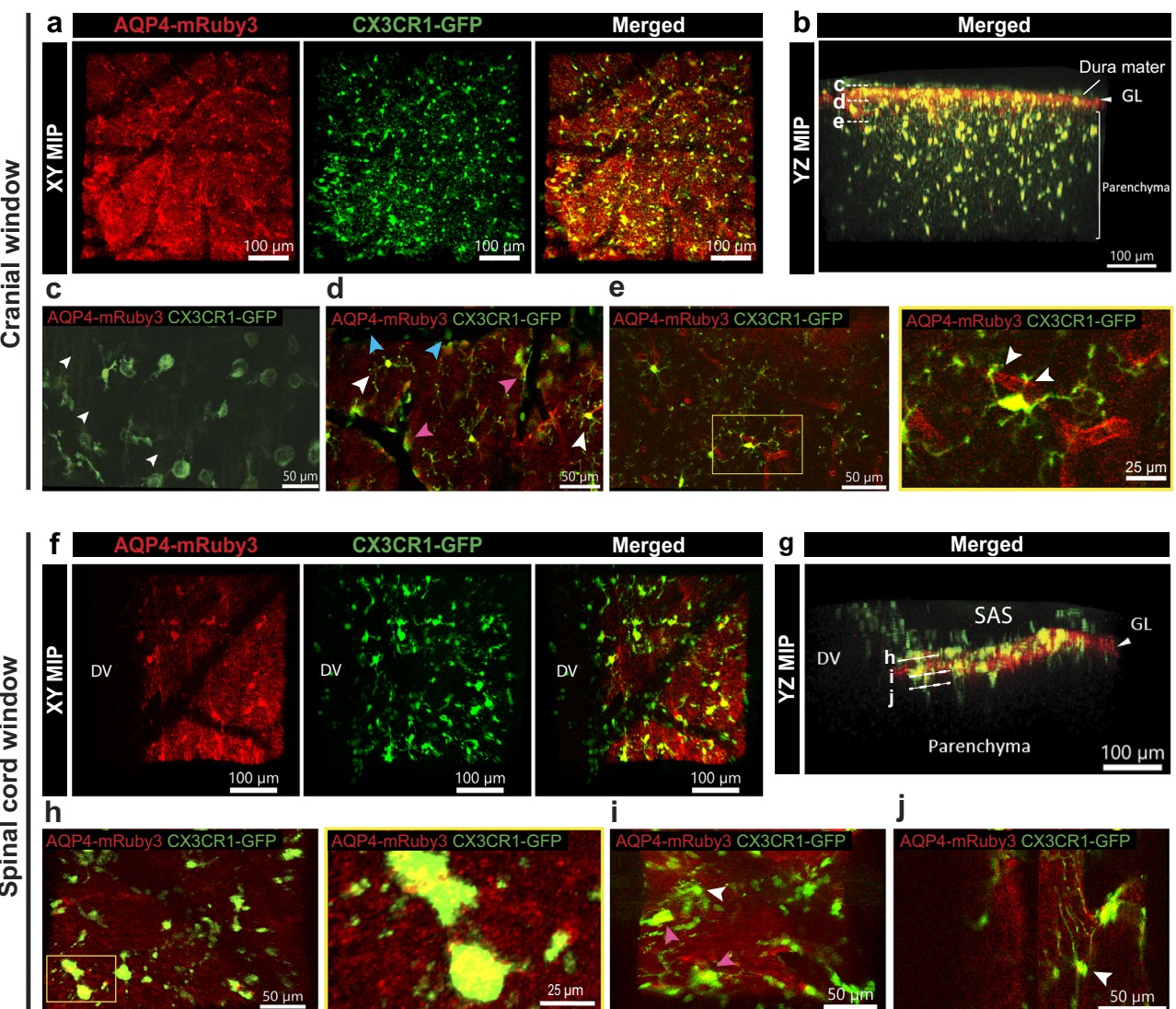

**Fig. 7 | Visualization of the glia limitans in Aqp4-mRuby3 mice allows to distinguish CX3CR1-GFP⁺ BAMs and microglia in vivo. a–e** Representative images of cranial window preparation in CX3CR1-GFP; Aqp4-mRuby3 mice (n = 7). **a** XY MIP of the 2P-IVM of the cranial window showing the view from the top. The glia limitans (GL) is visible in red, the microglia and border-associated macrophages (BAMs) in green. **b** YZ MIP of the 2P-IVM showing lateral view from the field of view (FOV) in (**a**). The superficial GL separates the top compartment, including collapsed subarachnoid space (SAS) and dura mater, from the bottom compartment corresponding to the brain parenchyma. **c–e** 2P-IVM images depicting BAMs and microglia in optical sections (2 μm) at the levels shown in (**b**). **c** Distinct morphology of BAMs (green) resting on dura mater collagen fibers observed by faint signal from the second harmonic generation (SHG) in green (white arrowheads). **d** Spatial localization of BAMs (pink arrowheads) in the perivascular region, microglia (white arrowheads) in the parenchyma, and circulating monocytes (light blue arrowheads)

within vessel lumens. **e** Left, distribution of brain microglia in relation to penetrating vessels. Right, the yellow inset shows thin microglial processes contacting the perivascular GL (white arrowheads). **f–j** Representative images of spinal cord window preparation in CX3CR1-GFP; Aqp4-mRuby3 mice (n = 6). **f** XY MIP showing an overview of the spinal cord CX3CR1-GFP⁺ resident myeloid cells (green) in area adjacent to the dorsal vein (DV). **g** YZ MIP from the FOV in (**f**) shows the SAS and parenchyma defined by the GL. **h–j** 2P-IVM images showing the position of BAMs and microglia in the spinal cord in the respective optical sections (2μm) indicated in (**g**). **h** Left, distribution of superficial BAMs resting on the spinal cord GL. Right, the yellow inset shows the ameboid shape of BAMs. **i** Spatial localization of BAMs (pink arrowhead) and microglia (white arrowhead) in the spinal cord. **j** Few microglia (white arrowhead) are visible in the spinal cord below the GL. Images in this figure were acquired with λex = 1000 nm. SAS subarachnoid space, GL glia limitans, DV dorsal vein, MIP maximum intensity projection.

simultaneous excitation of the AQP4-Ruby3 and CX3CR1-GFP signals at λ=1000 nm. These dural macrophages remained largely immobile during our 1-hour video acquisitions (Supplementary Movie 8). Displaying the data in 2 μm optical sections allowed for simultaneous identification of CX3CR1-GFP⁺ perivascular BAMs, microglia and circulating monocytes in vivo (Fig. 7d–e). Perivascular BAMs were seen along leptomeningeal vessels in close proximity to the superficial glia limitans, corresponding to regions where the SAS is not collapsed around the brain. Microglia were often located close to perivascular positions, either extending their processes towards the perivascular glia limitans or directly ensheathing the glia limitans with their cell

bodies. Visualization of microglial fine processes was achieved in optical sections rather than in YZ maximum intensity projections (MIPs) (Fig. 7e and Supplementary Movie 7), allowing imaging of their dynamic scanning of the surroundings and their contacts with perivascular glia limitans in steady state conditions (Supplementary Movie 8).

2P-IVM imaging of the spinal cord of Aqp4-mRuby3; CX3CR1-GFP mice revealed the presence of numerous CX3CR1-GFP⁺ BAMs in the SAS, localized above the AQP4-mRuby3 signal from the superficial glia limitans. Additionally, BAMs were covering the outside of the dorsal vein and other leptomeningeal vessels (Fig. 7f–i and Supplementary

Movie 9). Their morphology was rather amoeboid-like and they remained mostly static in steady state conditions, with occasional local movements observed (Supplementary Movie 8). However, in contrast to the observations from the brain, only a few spinal cord microglia were visible both in the regions around the dorsal vein and in more lateral fields of view (FOVs) (Fig. 7g, j). Our observations highlight the lower numbers of microglia in the spinal cord white matter when compared to the cortical brain matter as previously suggested (summarized in ref. 29).

### The Aqp4-mRuby3 reporter mouse allows for in vivo imaging of the barrier functions of the glia limitans

The barrier properties of the glia limitans for passage of soluble mediators from the CSF into the CNS parenchyma remain to be explored[30]. To investigate if Aqp4-mRuby3 mice allow for imaging the barrier properties of the glia limitans for soluble CSF tracers we performed 2P-IVM of the brain of Aqp4-mRuby3 mice through thinned skull preparations. Injection of an AF633 conjugated anti-endoglin antibody via a carotid artery catheter allowed for visualization of the luminal vascular walls (Fig. 8a, Supplementary Fig. 7). Cisterna magna infused 10 kDa FITC dextran was observed along meningeal arteries in paravascular spaces at 24-50 minutes after infusion (Fig. 8a, Supplementary Fig. 7). The AQP4-mRuby3 signal highlighted that the 10 kDa FITC dextran remained restricted to the SAS above the superficial glia limitans. Furthermore, we did not observe any 10 kDa FITC dextran entering the periarterial space, which questions the glymphatic concept proposing CSF entry into the parenchyma along periarterial spaces[31].

To further analyze the potential barrier function of the glia limitans for different sized tracers we next performed 2P-IVM imaging of the brain and spinal cord in CNS border reporter mice through skull thinning or cervical spinal cord window preparations during and after cisterna magna infusion of a mix of 10 kDa FITC-dextran and PEGylated Flash Red 1 μm beads. One hour after infusion both the fluorescent beads and tracer could be detected in the SAS of the brain (Fig. 8b, c). Two hours after injection, both beads and tracer were detected within the entire SAS but were not observed to cross the glia limitans (Fig. 8c), underscoring the barrier function of the glia limitans. When imaging the cervical spinal cord, we observed the arrival of fluorescent beads in the SAS at 30 minutes after cisterna magna infusion followed by detection of the fluorescent tracer (Fig. 8d–f). At 45 minutes after cisterna magna infusion, the beads and tracer were detected within the entire SAS (Fig. 8e, f). Neither the beads nor the 10 kDa dextran were observed to cross the glia limitans, confirming its barrier properties for both soluble and larger particulate tracers (Fig. 8g, h).

### CNS border reporter mice allow for simultaneous visualization of CD4 T cell interactions with the CNS borders during autoimmune neuroinflammation

Several studies have investigated the trafficking of CD4 T cells into the CNS using 2P-IVM[32]. However, the lack of appropriate landmarks has hampered interpretation of their precise localization in CNS border compartments (dura mater, SAS, subpial and perivascular spaces) versus the CNS parenchyma. To study CD4 T cell trafficking in vivo within the different CNS compartments we induced active EAE (aEAE) in female CNS border reporter mice that had either been adoptively transferred with naïve GFP+ 2D2 CD4 T cells or were instead systemically injected with Cell Tracker Deep Red labeled Th1 cells during the imaging session. 2P-IVM of the cervical spinal cord at disease peak (day 15-22 p.i.) of EAE was performed to follow their trafficking into the CNS (Fig. 9a). The fluorescent landmarks provided by VE-cadherin-GFP and AQP4-mRuby3, together with the SHG arising from the dura mater and subpial collagen type I fibers permitted assignment of CD4 T cells to the different CNS compartments in vivo, namely within the vasculature, in the CNS border compartments (including the subpial space,

SAS and dura mater) or within the CNS parenchyma (Fig. 9b–h, Supplementary Fig. 8, Supplementary Movie 10). Th1 cells were observed to sporadically cross the BBB and to migrate within the SAS (Supplementary Movie 11). GFP+ 2D2 CD4 T cells were assigned to the vasculature when located in the lumen of blood vessels outlined by endothelial VE-cadherin-GFP signal (Fig. 9b); to the SAS when positioned between the VE-cadherin-GFP+ layers marking the arachnoid and pial fibroblasts; to the subpial space when located between the VE-cadherin-GFP+ pia mater and the AQP4-mRuby3+ glia limitans and finally to the parenchyma when located below the AQP4-mRuby3+ glia limitans (Fig. 9b–f). In more severe cases of aEAE (score 2), we detected enlarged perivascular spaces around venules, in which GFP+ 2D2 CD4 T cells we observed to actively crawl but seemed confined to that compartment (Supplementary Movie 12). Moreover, the subpial compartment seemed enlarged in regions directly adjacent to large-caliber blood vessels such as the dorsal vein, which was associated with a thicker collagen fiber layer (Fig. 9f). Quantification of GFP+ 2D2 CD4 T cells showed the highest number of cells per FOV in the parenchyma and the SAS, and the lowest number in the dura mater irrespective of the number of naïve GFP+ 2D2 CD4 T cells originally injected prior to aEAE induction (Fig. 9g). This pattern is consistent with the described routes of invasion of encephalitogenic CD4 T cells in the spinal cord during aEAE in vivo, where T cells initially accumulate in the SAS after extravasating through leptomeningeal venules or veins and subsequently infiltrate the parenchyma by crossing the glia limitans after antigen re-encounter on antigen-presenting cells localized in the SAS[33].

Furthermore, tracking the dynamic behavior of the GFP+ 2D2 CD4 T cells in the different CNS compartments above and below the AQP4-mRuby3+ glia limitans showed a significantly larger displacement due to faster crawling speed of CD4 T cells within the spinal cord parenchyma when compared to CD4 T cells crawling in the SAS or subpial space above the AQP4-mRuby3+ glia limitans (Fig. 9h). Most CD4 T cells moved in the XY plane above and below the AQP4-mRuby3+ glia limitans and showed increased directionality in the spinal cord parenchyma which may be due to their migration along spinal cord white matter tracks. In any case, these studies underscore the impact of the respective CNS compartment on the migratory behavior of CD4 T cells. Complementary ex vivo T cell assignment to the respective CNS compartments was achieved by confocal imaging of brain cryosections of the border reporter mice at the peak of EAE counterstained or not for laminin (Fig. 9i, Supplementary Fig. 8d). Taken together, these data show the suitability of the CNS border reporter mice for distinguishing CD4 T cell dynamics within the different CNS compartments and thus assigning their behavior to CNS immune surveillance or neuroinflammation.

### Aqp4-mRuby3 reporter mice allow for imaging of CD8 T cell interactions with the glia limitans during CNS immune surveillance and neuroinflammation

Making use of ODC-OVA mice, which express ovalbumin (OVA) as a neo self-antigen in myelin-forming oligodendrocytes[34] and VE-cadherin-GFP reporter mice, we previously showed that in the absence of OVA expression in the CNS, activated OVA-specific CD8 T cells get access to the SAS and cross the pia mater to perform CNS immune surveillance, while in ODC-OVA mice OVA-specific CD8 T cells persist in the CNS parenchyma and cause autoimmune neuroinflammation[24,35]. To further understand the role of the glia limitans in regulating CD8 T cell trafficking into the CNS we made use of ODC-OVA mice crossed into female CNS border reporter and Aqp4-mRuby3 reporter mice or CNS border reporter mice as controls. Adoptive transfer of naïve GFP-expressing OVA-specific OT-I CD8 T cells was followed by peripheral infection with LCMV-OVA, allowing for peripheral activation of OVA-specific CD8 T cells and their migration into the CNS. To investigate the anatomical localization of the OVA-specific CD8 T cells in the brain and spinal cord

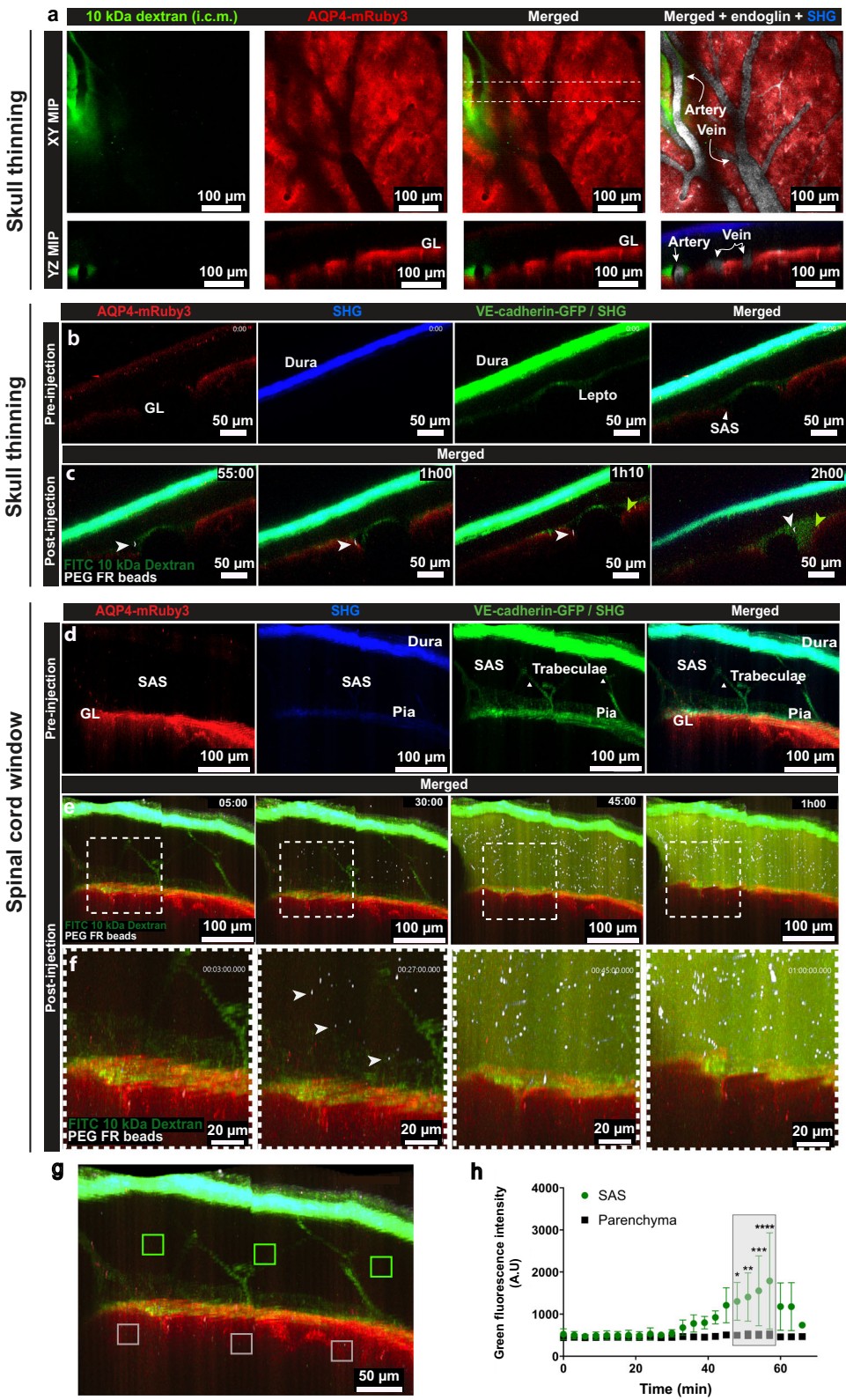

and their interaction with the glia limitans, we performed 2P-IVM at day 6-7 post-infection (p.i.) of the brain and spinal cord through cranial window, skull thinning and spinal cord window preparations (Supplementary Fig. 9, Fig. 10).

In the brain of female CNS border reporter mice and ODC-OVA; CNS border reporter mice the relative position of GFP+ CD8 T cells to AQP4-mRuby3 signal allowed for identifying CD8 T cells as parenchyma infiltrating or CNS surveilling T cells in the CNS border compartments (Supplementary Fig. 9). CD8 T cells above the AQP4-mRuby3 signal from the glia limitans were assigned to the border compartment, which encloses the SAS and subpial space. CD8 T cells located below the Aqp4-mRuby3 signal were identified as CNS parenchyma infiltrating cells. Whether CD8 T cells in the border compartment were located below the pia mater in the subpial space, or rather within the SAS on top of the pia mater remained uncertain, since the higher fluorescent intensity of the GFP+ CD8 T cells hindered the

**Fig. 8 | Visualization of CNS zonation in CNS border reporter mice in vivo.**
Representative 2P-IVM images of the brain surface through a thinned skull of Aqp4-mRuby3 (**a**) and CNS border reporter mice (**b, c**), and of the spinal cord in CNS border reporter mice (**d–g**). **a** During 2P-IVM 10 kDa-FITC dextran (green) was infused into the cisterna magna (i.c.m) and an anti-endoglin-AF633 antibody (white) injected via a carotid artery catheter to label the vasculature lumen. AQP4-mRuby3 depicting the glia limitans visible in red. Data were obtained by overlaying images with $\lambda ex_1 = 860$ nm and $\lambda ex_2 = 980$ nm. XY MIP at 24 minutes after i.c.m tracer infusion. YZ MIP from area between dashed lines in XY MIP. Data is representative of 2 mice. **b–h** A mix of 2.5 µL of 10 kDa dextran-FITC and PEGylated Flash Red beads (1 µm) was injected i.c.m at 1 µL/min during 2P-IVM. Time-lapse sequences were acquired for at least 60 min with $\lambda ex = 920$ nm and 1045 nm. SHG showing the dura mater and skull bone is visible in blue ($\lambda ex_1 = 920$ nm) or green ($\lambda ex_2 = 1045$ nm). **b, c** YZ MIP of a mouse skull thinning preparation, **b** before infusion, **c:** after infusion: visualization of beads (white arrowheads) and tracer (green arrowheads) arriving in the subarachnoid space (SAS) at indicated times. **d–f** YZ MIP of a spinal cord window preparation, **d** before infusion, **e** after infusion, visualization of the beads and tracer arriving in the SAS at indicated times. **f** Zoomed views from areas outlined in (**e**). **g, h** Longitudinal quantification of tracer mean fluorescence intensity (MFI) in the spinal cord. **g** shows volumes used for MFI measurement within the SAS (green squares) and parenchyma (white squares) in **e**. Graphs in (**h**) show the MFI of insets in **g**. Data from three mice were analyzed by a two-sided mixed-effects analysis with Šidák's multiple comparisons and are shown as mean ± SD. P values denoted as * = 0.0163, ** = 0.004, *** = 0.0004, ****<0.0001. Source data provided as a Source Data file, including exact p-values. SAS subarachnoid space, GL glia limitans, Lepto leptomeninges, MIP maximum intensity projection, i.c.m intra-cisterna magna, SHG second harmonic generation.

unambiguous detection of the delicate VE-cadherin-GFP signal from the pial fibroblasts.

2P-IVM of the spinal cord in female Aqp4-mRuby3 mice either injected with an Alexa Fluor (AF)−633-conjugated anti-endoglin antibody or an Alexa Fluor (AF)−680-conjugated 10 kDa dextran showed CD8 T cells in the CNS border compartments – this is above the AQP4-mRuby3 signal - performing CNS immune surveillance and few CD8 T cells in the parenchyma, below the glia limitans (Fig. 10b–d, h). This distribution was consistent with our previous in vivo results, where few CD8 T cells were detected below the pia mater, visualized by VE-cadherin-GFP expression[24].

Tracking the dynamic behavior of the CD8 T cells in the different CNS compartments above and below the AQP4-mRuby3$^+$ glia limitans showed, in accordance to our observations made for CD4 T cells (Fig. 9h), significantly longer, faster and straighter crawling trajectories for CD8 T cells in the spinal cord parenchyma than for CD8 T cells patrolling the CNS border compartments (Fig. 10i, j, k, Supplementary Movie 13).

During neuroinflammation, 2P-IVM of the spinal cord of CNS border reporter; ODC-OVA mice revealed a different immune landscape, with a significantly higher distribution of CD8 T cells along the glia limitans. There was a higher accumulation of parenchymal CD8 T cells (below the glia limitans) in comparison to CD8 T cells surveilling the border compartments (above the glia limitans) (Fig. 10e–g, h). Irrespective of their anatomical location, the CD8 T cells were found to be less motile, crawled at lower speed and in more random directions compared to CD8 T cells in the conditions of CNS immune surveillance (Fig. 10i–k, Supplementary Movie 14). CD8 T cells were also observed crossing the glia limitans into the CNS parenchyma (Supplementary Movie 15). These data therefore show that recognizing their cognate antigen in the CNS will reduce mobility of CD8 T cells along the CNS borders and allow for their migration across the glia limitans into the CNS parenchyma. Interestingly, some CNS parenchymal infiltrating CD8 T cells were observed to crawl in a longitudinal direction, parallel to the dorsal vein, suggesting they crawl along spinal cord white matter tracts (Supplementary Movie 14), similar to their behavior in the parenchyma during immune surveillance. In fact, parenchymal CD8 T cells showed even higher directionality as compared to CD4 T cells (Figs. 9h, 10k).

Taken together, these data show in accordance to the observations for CD4 T cells the suitability of the Aqp4-mRuby3 reporter mice for distinguishing CD8 T cell dynamics above and below the glia limitans and thus assigning their behavior to CNS immune surveillance or neuroinflammation.

## Discussion
Proper communication of neurons localized in the central nervous system (CNS) parenchyma requires a homeostatic environment that is not readily compatible with immune cells patrolling the tissue to ensure CNS immune surveillance. The CNS has therefore developed a unique relationship with the immune system, called CNS immune privilege[4]. CNS immune privilege is established and maintained by the brain barriers that divide the CNS into distinct compartments that differ by their accessibility to immune mediators and immune cells[1,4]. In the steady state, immune cell entry is restricted to CNS border zones such as the subarachnoid space (SAS) or perivascular spaces to screen for pathogenic alterations. During neurological disorders and associated with clinical disease, immune cell infiltration into the CNS parenchyma is, however, readily observed[36]. Based on these observations, we have proposed that the brain barriers ensure CNS immune privilege by an overall architecture that resembles that of a medieval castle surrounded by a two-walled castle moat[1,5]. In this analogy, immune cell entry into the CNS parenchyma requires two sequential and differentially regulated steps, namely migration across the outer brain barrier (outer wall) into the SAS or perivascular space ("castle moat"), which only under pathological conditions, is followed by immune cell passage across the glia limitans ("inner wall") into the CNS parenchyma[6]. While the molecular mechanisms involved in the multi-step extravasation across the BBB have been intensively investigated[37] little is known about the cellular and molecular mechanisms involved in immune cell trafficking across the glia limitans. The glia limitans is formed by subsets of astrocytes that are strategically positioned at the outer borders of the CNS parenchyma forming the superficial glia limitans[16,38] and by astrocytes extending foot-processes towards all blood vessels thus forming the perivascular glia limitans[15]. By prohibiting immune cell entry into the CNS parenchyma under steady state conditions the glia limitans therefore plays a key, yet underexplored, role in maintaining CNS immune privilege.

Studies in experimental autoimmune encephalomyelitis (EAE) as an animal model of multiple sclerosis have underscored that the molecular mechanisms mediating the migration of disease-inducing CD4 T cells across the BBB and the glia limitans are indeed distinct[39]. Elegant 2P-IVM studies have directly shown that in EAE, disease-inducing T cells cross the BBB and first reach the SAS or perivascular spaces, where recognition of their cognate antigens on border-associated macrophages (BAMs) or other border-associated antigen-presenting cells lead to their local reactivation and proliferation, which is prerequisite for their subsequent infiltration into the CNS parenchyma[32,40]. The Ag-specific reactivation and proliferation of T cells in the CNS border zones is accompanied by local secretion of the Th1 and Th17 signature cytokines, namely tumor necrosis factor (TNF-α)/interferon-γ (IFN-γ) and IL-17, respectively, which enhance activity of metalloproteinases MMP-2 and MMP-9 at the glia limitans[14]. Mice lacking MMP-2 and MMP-9 expression are completely resistant to clinical EAE, in the face of a massive accumulation of immune cells in the CNS perivascular spaces of brain vessels[14] underscoring the requisite role of MMP-2 and MMP-9 activity for immune cell invasion across the glia limitans. Interestingly, MMP-2 and MMP-9 were shown to cleave the astrocyte end-foot transmembrane extracellular matrix

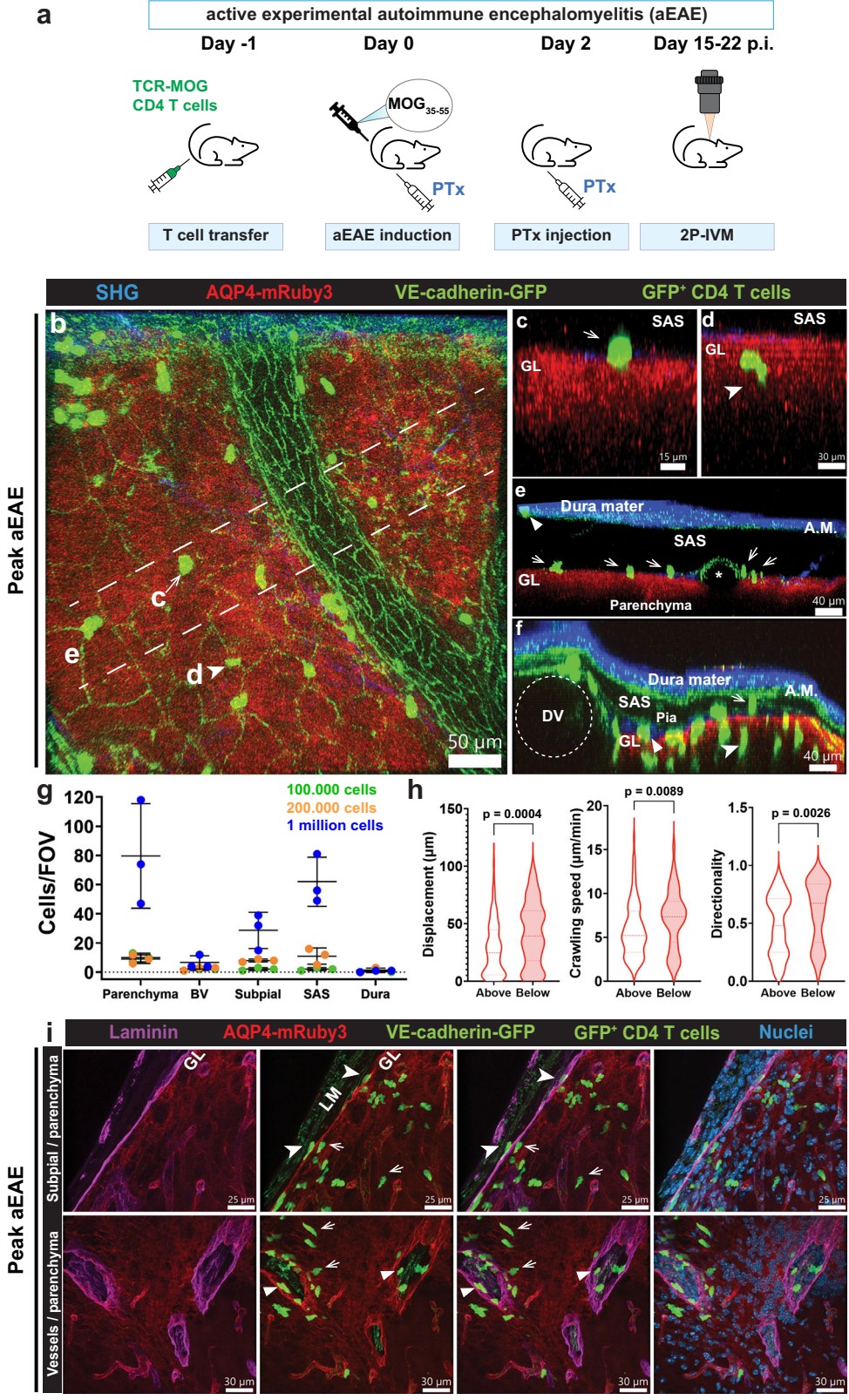

receptor β-dystroglycan, which results in the detachment of astrocyte endfeet from the parenchymal basement membrane, thus facilitating immune cell penetration across the glia limitans[14]. In addition, MMP-2 and MMP-9 activity modifies expression and activity of chemokines released at the glia limitans, thus promoting immune cell motility and exit out of the perivascular cuffs into CNS parenchyma[41]. While MMP-2 and MMP-9 activities thus seem to weaken the barrier properties of the glia limitans for immune cell entry, the observed upregulated expression of the tight junction molecules claudin-1 and claudin-4 and junctional cell adhesion molecule A (JAM-A) in reactive astrocytes of the glia limitans leads to the unusual formation of tight junction complexes[42]. Interestingly, while the tight junction sealing claudin-1 regulates water permeability across cellular barriers[43], claudin-4 rather modifies tight junctions permeability for ions and fluid clearance[44].

**Fig. 9 | CNS border reporter mice allow for precise assignment of 2D2 CD4 T cells to distinct CNS compartments during EAE. a** Female CNS border reporter mice received an intravenous injection of $1 \times 10^5$, $2 \times 10^5$, $0.5 \times 10^6$ or $1 \times 10^6$ naïve GFP⁺ 2D2 CD4 T cells at day −1 prior to induction of aEAE at day 0. 2P-IVM of the spinal cord was performed at peak of aEAE (15-22 days p.i.). **b–f** Representative images of 2P-IVM of the cervical spinal cord of mice following the protocol in (**a**). Data from overlayed images acquired at 920 and 1045 nm. The AQP4-mRuby3⁺ glia limitans is visible in red, endothelial and leptomeningeal adherens junctions are visible in green. 2D2 CD4 T cells are visualized by their GFP signal (green). SHG in blue from dura mater, trabeculae of the SAS and subpial collagen. **b–f** Overview of the spinal cord window of mice at peak of EAE (score 1) injected with $2·5 \times 10^5$ 2D2 CD4 T cells (n = 2). **c, d** YZ MIP zoom-in depicting localization of CD4 T cells from (**b**), above (**c**) or below (**d**) the glia limitans. **e** YZ MIP of region from (**b**) showing CD4 T cells in the dura mater (triangle) or the SAS (arrows). **f** YZ MIP of a FOV including the dorsal vein (DV), showing CD4 T cells in the subpial space (triangle) and in the parenchyma (arrowheads). **g** Quantification of CD4 T cells per FOV in the parenchyma, blood vessels, subpial space, SAS or dura mater. One mouse each was injected at day −1 with $1 \times 10^5$, $2 \times 10^5$, and $1 \times 10^6$ GFP⁺ 2D2 CD4 T cells, respectively. Data imaged at day 16-18 p.i. are shown for each mouse, from 1 aEAE induction, as mean ± SD, where each point represents a FOV (n = 1). **h** Violin plots showing displacement, crawling speed, and directionality of CD4 T cells during the observation period of 10 minutes. A minimum of 30 cells per condition were analyzed from 3 mice at the peak of aEAE (score 2, day 15-22 p.i.) injected at day −1 with $0.5 \times 10^6$ GFP⁺ 2D2 CD4 T cells, from 1 aEAE induction. Manual cell tracking was performed with Imaris 9.8 software. Data were analyzed using a two-sided unpaired parametric t-test. Source data are provided as a Source Data file. **i** Confocal imaging of 20 μm thick skull cryosections from CNS border reporter mice at the peak of aEAE (n = 3) counterstained for laminin (magenta). Top panel shows the surface of the brain, where cells are found in the subpial space (arrowheads) or parenchyma (arrow). Bottom panel shows CD4 T cells within the CNS parenchyma (arrow) or in the perivascular space (triangle). PTx pertussis toxin, MIP maximum intensity projection, A.M arachnoid mater, SAS subarachnoid space, GL glia limitans.

The observation that astrocyte specific-deletion of claudin-4 exacerbated EAE[42] underscores that even under neuroinflammatory conditions tight junctions newly formed by glia limitans astrocytes are critical for maintaining CNS compartmentalization and for controlling fluid equilibrium at this border zone. In addition to contributing to tight junction function, JAM-A was shown to mediate immune cell trafficking by homophilic interactions with JAM-A or by engaging the integrin leukocyte function adhesion molecule −1 (LFA-1) on immune cells. Mice deficient for JAM-A develop ameliorated EAE characterized by T-cell trapping in CNS border zones[45] suggesting that JAM-A rather than mediating immune cell entry across the BBB, is involved in their migration across the glia limitans. This is underscored by the observation that astrocyte-specific deletion of JAM-A ameliorated clinical EAE by reducing expression of MMP-2 and T-cell infiltration into the CNS parenchyma[46]. Taken together, these findings underscore the essential and most importantly active role of the glia limitans as a brain barrier that is readily reinforced during neuroinflammation to maintain control over fluid, immune cell and immune mediator exchange between the periphery and the CNS parenchyma.

While the role of astrocytes as modulators of CNS immunity has been appreciated, their active role as effectors in CNS immunity is emerging only recently[47]. Astrocytes within the CNS are very heterogenous and different astrocyte subtypes are defined largely based on their phenotypes and transcriptional signatures in addition to their anatomical location within the CNS[47]. Despite their strategically important localization at the CNS parenchymal border, the gene expression profile and nature of specific functional contributions of astrocytes forming the perivascular and superficial glia limitans to CNS health remain largely unknown. Interestingly, a recent report describes expression of myocilin specifically in glia limitans forming astrocytes in mice and humans[38]. While the precise function of myocilin remains to be explored, this finding underscores that the highly specialized subset of glia limitans forming astrocytes is evolutionarily conserved from rodents to humans and thus likely plays an important role in maintaining CNS homeostasis and CNS immune privilege.

In vivo imaging of the CNS at the single cell level has greatly advanced our understanding of afferent and efferent pathways of CNS immunity[32,48]. However, in most of these studies simultaneous imaging of immune cells and immune mediators and the respective brain barriers is lacking. Therefore, such studies fail to assign immune cell and immune mediator localization to a specific CNS compartment, which is, however, essential to understand the different mechanisms of immune cell entry and function underlying CNS immune surveillance versus CNS pathology. Making use of VE-cadherin-GFP knock-in mice that were originally created to image vascular endothelial adherens junctions[49], we recently found by 2P-IVM that VE-cadherin-GFP also illuminates the adherens junctions of the cells forming the arachnoid and pia mater in the brain and spinal cord of mice[10,24], which allowed

for imaging the barrier properties of the arachnoid and pia mater throughout the brain and spinal cord in health and disease[24]. This previous study furthermore showed that T cells crawl on the pia mater fibroblasts and mainly cross this leptomeningeal layer under conditions of neuroinflammation[24]. Lack of simultaneous visualization of the glia limitans in that previous study did, however, not allow for determining whether T cells remained in the subpial space or readily crossed the glia limitans to reach the CNS parenchyma. To allow visualization of the barrier properties of the glia limitans, we here show the development and use of a fluorescent reporter mouse for the superficial and perivascular glia limitans. To this end, we took advantage of the well documented highly enriched and polarized expression of the water channel aquaporin-4 (AQP4) in astrocyte subsets forming the superficial and perivascular glia limitans[19,50] and developed an Aqp4-mRuby3 knock-in reporter mouse. Aquaporins are a family of small integral membrane transport proteins facilitating bi-directional water movement across cell membranes in response to osmotic gradients (summarized in ref. 51). AQP4 is expressed as a short (M23) and long (M1) isoform with translation initiation sites at Met-23 and Met-1, respectively[52]. While each monomer contains a separate water pore, AQP4, like all aquaporins, forms stable tetramers in membranes[53]. Furthermore, plasma membrane assembly of AQP4 includes higher-order complexes referred to as orthogonal arrays of particles (OAPs)[20]. Thus, insertion of a fluorescent protein in the AQP4 molecule required careful consideration to avoid any impact of the fusion protein structure on function. Therefore, we specifically chose mRuby3 as a monomeric, bright and photostable red fluorescent protein[54] for insertion into the *Aqp4* locus. Our construct design was based on a previous thorough in vitro analysis of C- and N-terminal and extracellular loop insertions of GFP into AQP4 and their respective impact on OAP assembly[20]. As N-terminal GFP insertion into AQP4 was found to interfere with OAP assembly, we decided to insert mRuby3 into the second extracellular loop, exposing mRuby3 on the astrocyte cell surface.

Aqp4-mRuby3 reporter mice showed correct localization[22] of the AQP4-mRuby3 fusion protein in tissue sections ex vivo, as well as in vivo as observed by 2P-IVM. Importantly, Aqp4-mRuby3 was highly polarized in the endfeet of astrocytes forming the perivascular glia limitans and more diffusely distributed in astrocytes forming the superficial glia limitans. Exploring the morphology of the superficial glia limitans at the ultrastructural level we here confirm previous observations[16,17] that the superficial glia limitans is formed by astrocyte cell bodies extending long cellular processes as previously described. Importantly, the AQP4-mRuby3 protein was nevertheless polarized towards the outer surface of the surface astrocytes, forming the superficial glia limitans, allowing for a continuous polarized AQP4-mRuby expression at the transition zone from the superficial to the perivascular glia limitans. Our Aqp4-mRuby3 reporter mice also

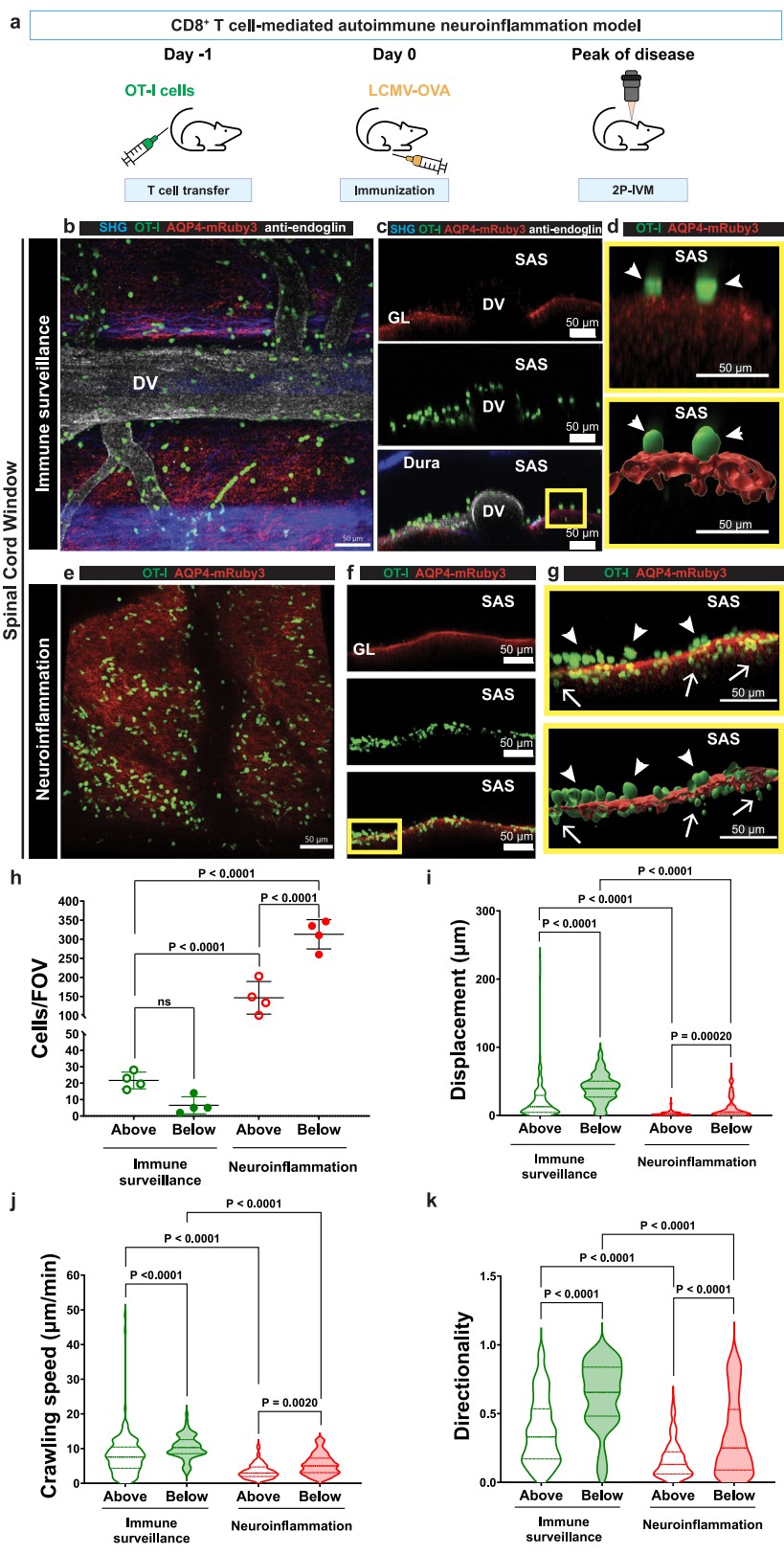

recapitulated the previously reported loss of polarized localization of AQP4 in astrocyte endfeet surrounding inflamed microvessels during EAE[22]. In accordance with this previous study, we here observed AQP4-mRuby3 to redistribute over the cell surface of perivascular astrocytes thus making this reporter mouse also suitable to study the involvement of AQP4 in edema formation in the CNS. Finally, lack of astrocyte foot swelling at the ultrastructural level and lack of alterations in water

content in the CNS of heterozygous Aqp4-mRuby3 reporter mice during health and EAE underscore their suitability for imaging the barrier function and thus functional contribution of the glia limitans to CNS zonation in vivo.

Crossbreeding the Aqp4-mRuby3 reporter with the previously described VE-cadherin-GFP reporter mice allowed us to obtain a border reporter mouse suitable for concurrent 2P-IVM imaging of the dura

**Fig. 10 | Aqp4-mRuby3 reporter mice allow distinction between CNS border patrolling from CNS infiltrating CD8 T cells in CNS immune surveillance and neuroinflammation. a** Female mice received an intravenous injection of $2 \times 10^5$ naïve GFP$^+$ OT-I T cells followed one day later by a peripheral infection with LCMV-OVA. On day 7 after infection, 2P-IVM of cervical spinal cord window was performed. Aqp4-mRuby3 reporter mice (**b–d**) or CNS border reporter; ODC-OVA mice (**e–g**) were used. **b–g** Representative XY MIP (**b, e**) and YZ MIP (**c, f**) images of the spinal cord surface are shown. CD8 T cells (green) can be distinguished above the glia limitans (arrowheads) and below (arrows). 4 mice were analyzed per condition from 1 CD8 T cell-mediated model induction. **d, g** YZ MIP zoom-ins of yellow boxed areas in **c, f** showing the fluorescence signals (top panel) and their segmented surfaces (bottom panel) rendered with Imaris 9.8 software. **b–d** Data from overlaying images acquired with 2P excitation wavelengths of 980 nm and 860 nm. The dura mater and subpial collagen are visible in blue due to SHG of the collagen type I fibers. The glia limitans is visible in red, vessels were labeled by intra-carotid injection of an Alexa Fluor (AF)−633-conjugated anti-endoglin antibody (white), GFP$^+$ OT-I CD8 T cells are visible in green. **e–g** Data acquired with 2P excitation wavelengths of 920 nm and 1045 nm. The glia limitans is visible in red. **h** Quantification of cells per FOV above or below the glia limitans during immune surveillance (**b–d**) or neuroinflammation (**e–g**) conditions. Data representative of 4 mice are shown as mean ± SD and were analyzed using a two-way ANOVA with Fisher´s LSD multiple comparisons. **i–k** Violin plots of the displacement (**i**), crawling speed (**j**), and directionality (**k**) of OT-I CD8 T cells. A minimum of 30 cells per compartment were analyzed from 4 CNS border reporter (immune surveillance) and 3 CNS border reporter; ODC-OVA mice (neuroinflammation). Manual cell tracking was performed with Imaris 9.8 software. Data were analyzed using a two-way ANOVA with Fisher´s LSD multiple comparisons. Source data are provided as a Source Data file. DV dorsal vein, MIP maximum intensity projection, SHG second harmonic generation, SAS subarachnoid space, GL glia limitans.

mater with its high collagen type I content (second harmonic generation (SHG)), the leptomeninges (VE-cadherin-GFP), the subpial space with its collagen bundles (SHG) and the glia limitans (AQP4-mRuby3) on the brain and spinal cord surface. Vascular endothelial cell as opposed to fibroblast VE-cadherin-GFP signal can be distinguished by the linear versus punctate appearance of the labeled adherens junctions as previously described[24]. This border reporter mouse allows for simultaneous imaging of the glia limitans and the vascular and leptomeningeal borders and thus for direct visualization of the CNS zonation established for CNS immune surveillance.

Creating Aqp4-mRuby3; CX3CR1-GFP double heterozygous reporter mice allowed to distinguish the CX3CR1-GFP expressing resident myeloid subsets, namely BAMs and microglial cells, specifically based on their localization with respect to the glia limitans in vivo. Previous studies relied on identifying microglial cells as distinct from BAMS merely based on their ramified cell shape (summarized in ref. [55]) or based on the calculated imaging depth during 2P-IVM[26]. Surgical window preparations can induce injury, leading to activation of BAMs and microglia and thus changes in cell shape in addition to inducing edema at the CNS surface. Thus, alterations in imaging distances while lacking a visual landmark for the glia limitans may lead to incorrect assignments of BAM and microglial behavior. Our current study further confirmed the recently described intimate contacts of microglial processes extending across the glia limitans towards endothelial cells forming the BBB[56]. Thus, the Aqp4-mRuby3 reporter mouse will be highly useful to study the exact mechanisms by which microglia as compared to BAMs contribute to neuroimmune interactions.

The barrier properties of the glia limitans for passage of soluble mediators from the CSF into the CNS parenchyma are a matter of debate[57,58]. In this context, the glymphatic hypothesis has obtained a lot of attention due to its potential impact on the understanding of CNS pathologies and functions on sleep. Following the distribution of different fluorescent tracers injected into the cisterna magna of mice, the glymphatic concept proposes active movement of CSF along periarterial spaces, followed by AQP4-mediated facilitated convective transport of CSF through the brain parenchyma and finally CSF exit via perivenous spaces[31]. Making use of the Aqp4-mRuby3 reporter mice, we here show that neither cisterna magna injected 10 kDa FITC-dextran nor PEGylated Flash Red 1 μm sized beads crossed the glia limitans. Neither of the fluorescent tracers was observed to enter periarterial spaces extending into the CNS parenchyma as proposed by the glymphatic hypothesis[24,58,59]. Periarterial tracer influx as proposed by the glymphatic hypothesis was shown to increase under conditions of anesthesia. A detailed study comparing the impact of different anesthesia protocols on periarterial CSF tracer distribution in mice identified the highest periarterial tracer influx under those conditions producing the lowest heart rates[60]. Intermediate periarterial tracer influx in that study was, however, observed at the heart rates maintained with our protocol, where anesthesia was induced by an initial injection of an anesthesia cocktail prior to surgery followed by low-dose isoflurane (0.5% to 1%) during CSF imaging. Inhibition of CSF flow has indeed been observed with high doses of isoflurane (2-3%)[59,61,62]. However, studies directly comparing high and low doses of isoflurane anesthesia showed that high isoflurane doses (2-3%) also prohibited cisterna magna injected tracer movement into the cortical SAS[59]. Thus, although the precise effects of the anesthesia on CSF dynamics in our experiments are not known, the used low-dose isoflurane protocol and the observed heart rate limits the previously reported detrimental effects on CSF flow with this anesthetic.

Taken together, both the Aqp4-mRuby3 reporter mice and the border reporter mice allowing to visualize the glia limitans and the glia limitans with the leptomeningeal and vascular borders will therefore be instrumental in clarifying the fluid dynamics in the CNS during health and disease, as well as sleep and awake states.

Finally, when employing CD4 and CD8 T cell-mediated autoimmune neuroinflammation models, our study shows that the border reporter mice allow for in vivo imaging of the dynamic CD8 and CD4 T cell interactions with the leptomeningeal layers, the glia limitans, and within the parenchyma, including their quantification and precise characterization. At the peak of neuroinflammation, most CD4 T cells were found in the SAS and parenchyma, recapitulating the known CNS invasion routes of autoaggressive CD4 T cells in this animal model. Remarkably, the lowest number of T cells was found in the dura mater, questioning the involvement of this border compartment at this stage of neuroinflammation. While during CNS immune surveillance CD8 T cells crawled in random directions above the glia limitans, during neuroinflammation their motility in CNS border compartments and the CNS parenchyma was reduced. Individual T cells were observed crossing the AQP4-mRuby3$^+$ glia limitans. Interestingly, the crawling speed and the directionality of crawling of both, CD4 and CD8 T cells were higher in the T cells below the glia limitans suggesting that once these T cells have crossed the glia limitans they migrate along the spinal cord white matter tracks. Furthermore, the border reporter mouse also allowed us to observe that in perivascular cuffs, which are a typical pathological hallmark of EAE and multiple sclerosis, T cells actively crawl within the confines of the perivascular space. Our in vivo imaging data using the CNS border reporter mice thus allows to observe different migratory behaviors of CD4 T cells in the different CNS compartments which allows to assign their specific behaviors to CNS immune surveillance versus neuroinflammation.

A remaining shortcoming of our study is that we cannot clarify if extracellular loop insertion of mRuby3 affects AQP4 function in vivo. Homozygous Aqp4-mRuby3 mice displayed a small but significant increase in brain but not spinal cord water content when compared to heterozygous Aqp4-mRuby3 and WT littermates suggesting that lack of expression of WT AQP4 leads to impaired water balance specifically in the brain. If this is due to steric hindrance mediated by the extracellular mRuby3 impacting the formation of higher order membrane structures

of AQP4-mRuby3 or if the individual Aqp4-mRuby3 molecule is impaired in water transport capacity cannot be easily solved in vivo as even AQP-deficient mice do not display significant changes in overall brain structure, vascular anatomy, intracranial pressure and BBB function (summarized in ref. 63). It was, however, observed that AQP4-deficient mice develop ameliorated EAE and neuroinflammation upon intrathecal LPS injection[64]. Heterozygous Aqp4-mRuby3 reporter mice did not display any altered phenotype with respect to brain and spinal cord water balance and development of EAE. Thus, investigating EAE pathogenesis in homozygous AQP4-mRuby3 reporter mice may allow to better understand the function of AQP4-mRuby3 at the molecular level. As the focus of this study was to develop a glia limitans fluorescent reporter, we decided against pursuing these experiments according to the 3 R rules of animal experimentation. Taken together as the water content of the brain and spinal cord was not altered in heterozygous Aqp4-mRuby3 mice and we also failed to see any significant differences of astrocyte end-foot anatomy at the ultrastructural level and no significant differences in EAE development between Aqp4-mRuby3 mice and wild-type littermates, we consider the heterozygous Aqp4-mRuby3 mice as suitable and reliable for the imaging of barrier properties of the glia limitans in health and disease.

Our present study provides direct evidence for barrier properties of the glia limitans to soluble tracers and immune cells in vivo. In general, the Aqp4-mRuby3 reporter mice by themselves or crossed with yet other cellular reporters and/or mice deficient in specific molecules will allow to address open questions on the role of the brain barriers in CNS immunity and pathology.

## Methods
### Mice
Male and female mice between 8 and 12 weeks were used in this study. The Aqp4-mRuby3 reporter mouse line was developed in collaboration with Polygene AG (Rümlang; Switzerland), as described below, in a C57BL/6 N genetic background and also backcrossed to C57BL/6 J. C57BL/6 J and C57BL/6 N wild-type mice were obtained from Janvier (Genest Saint Isle, France). VE-cadherin-GFP knock-in mice expressing a C-terminal GFP fusion protein of VE-cadherin in the endogenous VE-cadherin locus *(Cdh5tm9Dvst)*, allowing for imaging of the endothelial and meningeal adherens junctions, were described before[24,49]. CX3CR1[GFP] knock-in mice (B6.129P2(Cg)-*Cx3cr1*[tm1Litt]/J) allowing for imaging of CNS myeloid cell subsets by expression of EGFP were described before[27]. OT-I mice (C57BL/6JTg(Tcra/Tcrb)1000Mjb) contain a transgenic TCR, recognizing OVA residues 257-264 (SIINFEKL) in the context of H2-K[b][34,65]. OT-I GFP mice expressing GFP ubiquitously were generated by crossing OT-I mice with UBI-GFP mice (Tg(UBC-GFP)30Scha)[66]. ODC-OVA mice expressing ovalbumin specifically in oligodendrocytes have been described before[34]. VE-cadherin-GFP ODC-OVA mice were generated by crossing ODC-OVA mice with the VE-cadherin-GFP knock-in mice as described before[24]. 2D2 mice (C57BL/6-Tg(Tcra2D2, Tcrb2D2)1Kuch/J) contain a transgenic TCR recognizing MOG residues 35-55 in the context of MHC class II (I-A[b])[67]. 2D2 GFP mice expressing GFP ubiquitously were generated by crossing 2D2 mice with UBI-GFP mice (Tg(UBC-GFP)30Scha)[66]. All mice were housed in individually ventilated cages under specific pathogen-free conditions at 22 °C with a 13:11 h light:dark cycle and fed *ad libitum* with a chow diet and water. Animal procedures were approved by the Veterinary Office of the Canton of Bern (permits no. BE77/2018, BE73/2021, BE31/2017, BE98/2020 and BE113/2023) and according to institutional and standard protocols for the care and use of laboratory animals in Switzerland.

### Production of Aqp4-mRuby3 knock-in reporter mouse line
**Targeting Strategy.** A gene-targeting approach by homologous recombination in embryonic stem cells was pursued for the introduction of mRuby3 into exon 2 of the *Aqp4* gene. In the final gene-

modified allele, the mRuby3 cassette containing exon 2 is flanked in introns 1 and 2 by loxP and lox2272 sites, respectively, to facilitate subsequent recombinase-mediated cassette exchange (RMCE) to allow for exchange of the modified exon 2 by different fluorescent fusion designs. Suitable positions for insertion of the flippase (FLP) recombination target (FRT)- and lox sites were identified by comparing the intronic sequence upstream and downstream of exon 2 of different species. Alignment of mouse, rat, and human intronic regions revealed less conserved positions that were chosen as regions for FRT and lox site insertions. A targeting construct was produced (Supplementary Fig. 1), in which a linker-flanked monomeric fluorescent protein mRuby3 cassette was inserted in between valines 141 and 142 of the exon 2 of the *Aqp4* gene (Fig. 1a). An FRT-flanked neo cassette was inserted into intron 1 (Fig. 1a). Also, LoxP and lox 2272 sites were placed downstream of the second FRT site in intron 1 and in intron 2, respectively (Fig. 1a; Supplementary Fig. 1) to allow recombinase-mediated cassette exchange (RMCE) of mRuby3 with other inserts coding for different fluorescent protein fusion variants if required.

**Targeting vector construction.** For the insertion of the fragment containing the mRuby3 cassette within exon 2 of the *Aqp4* gene described below, a targeting vector based on pUC57 was generated containing a 2.5 kb short arm (SA) and 4.9 kb long arm (LA) of homology. These homology arms were amplified by PCR from C57BL/6 N genomic DNA. A 1571 bp long DNA fragment was synthesized that contains mRuby3 already inserted in its position in exon 2, as well as introns 1 and 2 from the upstream insertion point up to the downstream insertion point. This fragment also contained the downstream lox2272 site and all required unique sites for further cloning and an additional EcoRI site for screening by Southern blotting. The desired flexible connection between the *Aqp4* sequence coding for the second extracellular loop and the mRuby3 polypeptide was provided by a linker of a 3-fold GGGS pattern (3xGGGS; G = glycine; S = serine). This linker was described[68] and successfully used in several constructs of previous Polygene research projects. A fragment containing the FRT-flanked neo-casstte followed by a LoxP site was inserted upstream of exon 2. The integrity of the completed targeting vector was confirmed by restriction enzyme analysis and sequencing. This construct was then electroporated into C57BL/6 N embryonic stem (ES) cells and submitted to G418 selection. G418-resistant clones were tested as described below, and correctly targeted ES-cells were injected into blastocysts, which were then transferred to CD1 recipients. Chimeric founders were bred to FLP-deleter mice (hACTB-FLPe; B6g-Tg(ACTFLPe)9205Dym/NPG) to remove the FRT-flanked neo cassette. Their offspring were bred to C57BL/6N mice under specific and opportunistic pathogen-free (SOPF) conditions at the animal facility of the Theodor-Kocher Institute and, in addition, backcrossed to C57BL6/J for more than 10 generations.

**Transfection of embryonic stem cells.** For electroporation, 30 μg of targeting vector DNA was linearized using the restriction enzyme AscI, and electroporated into $1.7 \times 10^7$ or $2 \times 10^7$ C57BL6/N-derived ES cells. G418 at 0.2 mg/ml was used to select stably transfected clones. After 8 days of selection, a total of 10 times 96 clones were picked and analyzed by PCR analysis. PolyGene uses for BL/6 N ES cell injections a strain of mice that was based on CRL-delivered C57BL/6 N mice with a spontaneous recessive mutation causing dark gray fur color. The strain was carefully marker genotyped and is pure C57BL/6 N, but unlike the JAX-available albino Tyr strain is easily superovulatable and breeds well.

**Analysis of ES cell clones by PCR.** We used PCR-based and Southern blot approaches to screen and validate clones. 6 × 96 clones were isolated after selection with G418, and lysates were tested via screening PCR analysis. The following primers were used: 5′-GTTGTGCCC

AGTCATAGCCGAATAG-3' and 5'-CTCAGAAGACAGCACCTGTAATAGC-3'. The amplicon size was 3239 bp. PCR screening revealed 18 positive clones, which were expanded and their DNA prepared for further analyses.

**Confirmation of correct homologous recombination by Southern blot analysis.** To confirm correct targeting and homologous recombination with the *Aqp4* locus, Southern blot analysis was performed using EcoRI-digested genomic DNA of 18 clones. DNA was hybridized with a 3' external probe of 500 bp generated using the following primers: 5'-CCCATTTCCCAGAGACTACC-3', 5'-TAGTGCCGTTCCCAAGAGAC-3'. Southern blot analyses confirmed correct homologous recombination for 4 clones (6B6, 6C3, 9B1, 9F11), which were selected for blastocyst injection.

**Chimeric mice.** Two rounds of blastocyst injections with 3 different ES cell clones into C57BL6/g gray blastocysts were performed, which yielded a total of 15 chimeras. Chimeric mice were bred with wildtype C57BL/6g mice, and black offspring (i.e., mice with germ line transmission) were screened by PCR for the presence of the neo-cassette using the primer neo.MP1 (5'-GCTGTGCTCCACGTTGTCCAC-3') and neo.MP6 (5'-GGAGCGGCGATACCGTAAAG-3'). Neo-positive mice were bred with Flp-deleter mice to remove the FRT-flanked neo-cassette. Absence of the neo-cassette was determined by PCR using the primers I019.23 (5'-TGGCGATGGTTTGCTGAACACTC-3') and I019.24 (5'-CCTTTCTAGGGATGGTTGGATTG-3'), binding to endogenous sequences on either side of the neo cassette. All the correctly targeted neo-deleted mice are derivatives of the ES cell clone 9B1.

## Mouse models of autoimmune neuroinflammation

**Active EAE.** Active EAE (aEAE) was induced in 8–12-week-old female transgenic mice and WT littermates by immunization with 200 μg MOG$_{aa35-55}$-peptide in Complete Freund's Adjuvant (CFA, LabForce; Santa Cruz Biotechnology) exactly as described[9]. Pertussis toxin (List Biological Laboratories, Campbell, US), 300 ng per mouse, was applied intraperitoneally (i.p.) at day 0 and day 2 post-immunization. Weight and clinical severity were assessed twice daily and scored as follows: 0: asymptomatic; 0.5: limp tail; 1: hind leg weakness; 2: hind leg paraplegia; 3: hind leg paraplegia and incontinence (humane end-point). Mice were defined at pre-onset (day 10-11 p.i., clinical score 0), onset (day 12-14 p.i., clinical score 0.5 to 1), peak (day 15-22 p.i., clinical score 1 to 2), or at the chronic phase of the disease (day 25-30 p.i., clinical score 1 to 0.5).

**CD8 T-cell mediated CNS immune surveillance and neuroinflammation.** 8-12-week-old female mice were intravenously injected with 2×10$^5$ naïve OT-I GFP CD8$^+$ expressing T-cells. Briefly, CD8$^+$ naïve T-cells were isolated from peripheral lymph nodes and spleen of OT-I GFP C57BL/6 J mice by negative selection using magnetic beads (CD8$^+$ T-cell isolation kit, EasySep, STEMCELL Technologies) as previously described[24,35]. 24h later, animals were infected with 10$^5$ plaque-forming units (PFU), OVA-expressing lymphocytic choriomeningitis virus (LCMV-OVA). Animals in the ODC -OVA background were monitored daily for clinical symptoms and scored as previously described[35,69]. Live cell imaging of CD8 T-cell-mediated CNS immune surveillance (in control mice) and neuroinflammation (in ODC-OVA background) was studied 6-7 days post viral infection when ODC-OVA mice display clinical disease.

## T cell proliferation assay

WT and heterozygous Aqp4-mRuby3 mice suffering from aEAE at day 30 p.i. were perfused with 10 ml of PBS. Spleens were collected, and single-cell suspensions were prepared. After erythrocyte lysis with ammonium-chloride-potassium cells were filtered through a 100 μm nylon mesh and cell suspensions were seeded in a 96 well-plate at a density of 1×10$^6$ cells/ml and cultured for 48 hr in RPMI-1640 supplemented with 10% FBS (FBS Gold, Seraglob), 10 U/ml penicillin-streptomycin, 2 mM L-glutamine, 1% (v/v) non-essential amino acids, 1 mM sodium pyruvate, and 0.05 mM β-mercaptoethanol (Grogg Chemie AG) as negative control, with anti-CD3/CD28 antibodies (1 μg/ml each) as positive control of polyclonal T cell activation, and increasing concentrations of MOG$_{aa35-55}$- peptide (10, 20 and 40 μg/ml). Cells were labeled with 0.2 μCi of $^3$H-Thymidine (Thymidine, [Methyl 3H] NET 027W001MC, Perkin Elmer) per well during the last 16 hours of culture. Sixteen hours post labeling, cells were harvested on glass fiber filters and samples were measured in scintillation fluid (Ultima Gold™) using a beta-scintillation counter (Hidex 600 SL).

## Differentiation of Th1 cells

Naïve T cells were isolated from peripheral lymph nodes and spleens of 8-12 weeks old female 2D2-TCR-MOG mice sacrificed by CO$_2$ euthanasia. Lymphoid organs were first homogenized and filtered through a sterile 100 μm nylon mesh to obtain a single cell suspension. After erythrocyte lysis, CD4$^+$ T cells were isolated using negative selection magnetic beads (EasySep™ mouse CD4$^+$ T cell isolation kit, STEMCELL Technologies) according to the manufacturer's instructions. For T-helper-1 cell (Th1) differentiation, naïve T cells (5 × 10$^5$/ml) were stimulated for 6 days in presence of 40 Gy irradiated antigen presenting cells (5 × 10$^6$/ml) isolated from C57BL6/J mouse and MOG$_{35-55}$ peptide (20ug/ml). Cells were cultured in restimulation medium (RPMI-1640 supplemented with 10% FBS (FBS Gold, Seraglob), 10 U/ml penicillin-streptomycin, 2 mM L-glutamine, 1% (v/v) non-essential amino acids, 1 mM sodium pyruvate, and 0.05 mM β-mercaptoethanol (Grogg Chemie AG)) in the presence of recombinant mouse IL-12 (20 ng/ml) and 5% IL-2 supernatant (self-produced from IL-2 producing IL-2x63AgO Lymphoma cell line). On day 6, freshly activated live cells were isolated by Ficoll 1.077 A (Ficoll Paque Plus, GE Healthcare) density gradient centrifugation and restimulated over 3 days as described on the first day of culture. Activated T cells were collected by Ficoll density gradient. Freshly activated live 2D2 Th1 cells were resuspended in fresh restimulation medium containing 5% IL-2 cell supernatant and seeded in petri dishes at a concentration of 5 × 10$^5$/ml and incubated at 37 °C in 5% CO$_2$ for an additional 24 h.

## Isolation and injection of naïve 2D2 CD4 T cells

Naïve T cells were isolated from peripheral lymph nodes and spleen of 5-8 weeks old female 2D2-TCR-MOG; Ubi-GFP mice sacrificed by CO$_2$ euthanasia. Single cell suspensions of the lymphoid organs were filtered through a sterile 70 μm cell strainer. GFP$^+$ CD4 T cells were isolated using negative selection magnetic beads (EasySep™ mouse CD4$^+$ T cell isolation kit, STEMCELL Technologies) according to the manufacturer's instructions. 1 × 10$^5$, 2 × 10$^5$, 5 × 10$^5$ or 1 × 10$^6$ 2D2 CD4 T cells in 100 uL of saline solution were intravenously injected on the day before aEAE induction (day −1 p.i.).

## Tissue preparation for fluorescence imaging

**Preparation of decalcified head and vertebral columns.** To obtain brain and spinal cord sections in the correct context of the skull and vertebral column, mouse heads and vertebral columns were prepared and decalcified. To this end, mice were deeply anesthetized with isoflurane and transcardially perfused with 10 ml PBS, followed by 10 ml of 4% PFA/PBS. Mice were then decapitated, and the skin and muscles were removed from the whole heads using forceps and scissors. Teeth, eye lenses, and mandibles were removed to allow for subsequent cryosectioning of the tissue. The whole heads were fixed in 4% PFA overnight, then rinsed and decalcified in 14% EDTA (Sigma-Aldrich) for 7-10 days at room temperature (depending on the softness of the tissues), followed by cryoprotection in 30% sucrose for 3 days, as previously described[70]. The EDTA buffer was replaced with fresh 14% EDTA every two days. For whole vertebral column preparations, the skin was

removed, and the vertebral columns were separated from the rib cages. Vertebral columns were then fixed in 4% PFA/PBS overnight, decalcified in 14% EDTA for 7 days followed by cryoprotection in 30% sucrose for 3 days. The decalcified mouse samples were finally embedded in Tissue-Tek OCT compound, snap frozen in a dry ice-cooled 2-methylbutane bath, and stored at −80 °C. Cryosections were performed (10-20 μm) with a CryoStar Nx50 cryostat (Epredia).

**Preparation of brains and spinal cords.** Mice were perfused with 10 ml PBS (Gibco), followed by 10 ml of 1% or 4% PFA/PBS under deep isoflurane anesthesia (Attane, Piramal Healthcare) exactly as described[24]. Brains were harvested by carefully cutting open the skull at the midline, avoiding injury to the leptomeninges. To harvest the spinal cord, the individual vertebrae of the spinal column were carefully cut lengthwise to extract the spinal cord by laminectomy, paying attention not to peel off the leptomeningeal layers. Brains and spinal cords were embedded in Tissue-Tek OCT compound, frozen in a dry ice-cooled 2-methylbutane (M32631, Sigma Aldrich) bath, and stored at −80°C. 10-20 μm cryosections were cut with a CryoStar Nx50 cryostat (Epredia).

## Immunofluorescence staining of cryosections and image analysis

The tissue cryosections were rehydrated with DPBS (Gibco) and then blocking buffer was applied (10% goat serum, 0.1% Triton X-100 (Sigma-Aldrich) in PBS pH 7.4) during 20 min at RT, followed by one night incubation in the primary antibody solution (2% goat serum, 0.1% Triton X-100 in PBS pH 7.4, primary antibody) at 4 °C. After three washes with DPBS, cryosections were incubated for 2 h with the secondary antibody. Sections were rinsed with DPBS, stained with DAPI (1:3000), and rinsed three times with DPBS for 15 minutes. Sections were mounted with embedding medium Mowiol (Sigma-Aldrich, Steinheim, Germany) prior to image analysis.

Images were acquired with a LSM800 confocal microscope (Carl Zeiss, Oberkochen, Germany) or with an Axiozoom.V16 fluorescence microscope (Carl Zeiss, Oberkochen, Germany) both equipped with ZEN 2.6.76 software (Carl Zeiss, Oberkochen, Germany). Images were then processed with Imaris 9.8 software (Oxford Instruments, Oxfordshire, England) and FIJI 2.14.0 (ImageJ, National Institute of Health, Bethesda, USA). Quantitative analysis of perivascular cuffs in coronal brain cryosections of mice suffering from EAE included cuffs observed around vessels in cross sections and vessels in longitudinal orientation. Cuffs were defined as small if their diameter was below 100 μm (coronally oriented) or their length was below 200 μm (longitudinally oriented). They were classified as large if their diameter exceeded 100 μm (coronally oriented) or their length exceeded 200 μm (longitudinally oriented).

## Antibodies

Primary antibodies used were monoclonal rat-anti mouse CD45 (clone M1/90, produced in house, undiluted hybridoma cell culture supernatant), rabbit anti-AQP4 antibody (Millipore, AB2218, 10 μg/mL), rabbit anti-mouse pan laminin (Novus, NB300-144SS, 5 μg/mL), rabbit anti-mouse laminin 1 + 2 antibody (Abcam, ab7463, 5 μg/mL), goat anti-mouse podocalyxin (R&D Systems, AF1556, 2 μg/mL), and rabbit anti-glial fibrillary acidic protein (GFAP) antibody (DAKO, ZO334, 24 μg/mL) and rat-anti mouse CD31 antibody (clone Mec13.3, produced in house, undiluted hybridoma cell culture supernatant). Normal rabbit IgG (Invitrogen, 02-6102, 10 μg/mL), goat IgG (R&D Systems, AB-108-C, 3.3 μg/mL) and rat IgG2a (Pharmingen, 553927, 10 μg/mL) were used as controls. The secondary antibodies used were Alexa Fluor (AF)−647-conjugated goat anti-rabbit IgG (H + L) cross-adsorbed antibody (ThermoFisher Scientific, cat no A-21244, 4 μg/mL), Cyanine (Cy)5-conjugated donkey anti-rabbit IgG antibody (Jackson

ImmunoResearch, 711-175-152, 7.5 μg/mL), AF647-conjugated donkey anti-goat IgG polyclonal antibody (Jackson ImmunoResearch, 705-605-003, 7.5 μg/mL), AF647-conjugated donkey-anti rat IgG (Invitrogen, A78947, 6.7 μg/mL), AF488-conjugated donkey anti-rat IgG (Thermo-Fisher, A-21208, 6.7 μg/mL) and AF488-conjugated donkey anti-rabbit IgG antibody (Invitrogen, A32790, 6.7 μg/mL). For in vivo labeling of the vascular lumen during 2P-IVM, a monoclonal rat-anti mouse endoglin (clone MJ7/18, produced in house, 40 μg/mouse) antibody was conjugated with AF633 (Alexa Fluor™ 633 Protein Labeling Kit, Thermo Fischer Scientific, A20170). Further details about the antibodies used in this study are detailed in Supplementary Table 1.

## Transmission and serial block-face scanning electron microscopy

**Tissue preparation.** The protocol used was adapted from the protocol of Knott et al.[71]. 10-week-old mice were perfused transcardially with 10 ml PBS, followed by perfusion with 300 ml of 2.5% glutaraldehyde (Agar Scientific, Stansted, Essex, UK) and 2% paraformaldehyde (Merck, Darmstadt, Germany) in 0.1 M sodium-cacodylate buffer (Merck, Darmstadt, Germany) pH 7.4 at RT under deep isoflurane anesthesia. Two hours after perfusion, mice were decapitated, and brains were removed. 70μm thick vibratome (Leica, Wetzlar, Germany) sections were cut through the neocortex, the brainstem and the cerebellum. The sections were washed in sodium-cacodylate buffer (0.1 M, pH 7.4) three times for 5 min and overnight, and postfixed for 40 min with 1 % OsO4 (Electron Microscopy Sciences, Hatfield, USA) and 1.5% potassium ferrocyanide in 0,1 M sodium-cacodylate buffer at RT. Solution was then changed for 1% osmium tetroxide in 0,1 M sodium-cacodylate buffer at RT for 40 min and the sections were rinsed three times for 5 min with distilled water. Samples were dehydrated by subsequent incubation in 70, 80, 96% ethanol (Grogg, Bern, Switzerland) for 15 min at each step at RT and three steps in 100% ethanol (Merck, Darmstadt, Germany) for 10 min, and finally twice in acetone for 10 min (Merck, Darmstadt, Germany).

Sections were then placed in a mix 1:1 of acetone:epoxy resin (Epon, Sigma-Aldrich, Buchs, Switzerland) overnight. The next day, samples were embedded in 100% Epon and left at RT to allow the sections to sink down. Finally, the sections were transferred to an oven at 60 °C for 5 days.

For transmission electron microscopy imaging, sections were made with an ultramicrotome UC6 (Leica Microsystems, Vienna, Austria), first semithin sections (1 μm) for light microscopy which were stained with a solution of 0.5% toluidine blue O (Merck, Darmstadt, Germany), and then ultrathin sections (75 nm). The ultrathin sections were mounted on 200 mesh copper grids, and contrasted with Uranyl acetate (Electron Microscopy Sciences, Hatfield, USA) and lead citrate (Leica Microsystems, Vienna, Austria) with an Ultrostainer (Leica Microsystems, Vienna, Austria).

**Image acquisition, montage reconstruction and analysis.** Sections were imaged with a transmission electron microscope (Tecnai Spirit, Thermo Fisher Scientific) for astrocytic endfeet analysis. Image acquisition was performed as follows: for each experimental group (n = 3 WT and n = 2 Het mice), three regions of interest (ROI) were imaged as described above (the neocortex, cerebellum and brainstem), allowing for analysis of 20 microvessels in each brain region. Images were acquired using an Olympus-SIS Veleta CCD camera for the overviews, and then montages were acquired using the SerialEM software and a FEI Eagle CCD Camera with a magnification of x13'000, obtaining images in the MRC file format. Image processing was performed using the IMOD software package[72]. Specifically, montage reconstruction was performed within Etomo using the "Align Serial Sections / Blend Montages" pipeline.

Astrocyte end-foot ultrastructure was evaluated by an expert in a blinded fashion. For each montage of a blood vessel section, the astrocyte end-foot overall surface/volume ratio and cytoplasmic density was considered and a swelling score ranging from 0 to 2 was assigned. Small astrocyte endfeet with homogenous and relatively dense cytoplasm were considered normal and assigned to a swelling score of 0. Moderately enlarged astrocyte endfeet with cytosol appearing less dense were given a swelling score of 1. Finally, markedly enlarged astrocyte endfeet with cytosol appearing mostly empty and including empty-looking membranous compartments were assigned a swelling score of 2. This numerical categorization allowed for comparison of the degree of astrocytic endfeet swelling and ultrastructural abnormalities between genotypes and brain regions.

For analysis of the superficial glia limitans, three-dimensional (3D) ultrastructural images were obtained by serial block-face scanning electron microscopy (SBF-SEM) using a Quanta FEG 250 SEM (FEI, Eindhoven, The Netherlands) equipped with an in situ 3View2XP ultramicrotome (Gatan). Images were acquired in Low Vacuum mode 50 Pa. The acceleration voltage was 5.00 kV, and the pixel dwell time was set to 3 µs. Pixel size was 0.06 µm, with a section thickness of 200 nm. Image acquisition was performed using a backscattered electron detector optimized for SBF-SEM (Gatan). Image stacks were aligned, normalized, and denoised using non-linear anisotropic diffusion in IMOD[73]. The final image montage was done in ImageJ.

### Brain and spinal cord water content measurement

Water content of mouse brains and spinal cords was quantified as previously described[74,75]. Briefly, 10-week-old female mice were sacrificed by $CO_2$ euthanasia followed by decapitation prior to dissecting the brains and spinal cords. The wet weight (W) of each individual tissue was obtained immediately after organ extraction. The dry weight (D) was measured after 24 h and 48 h in a 95 °C dry heated chamber. The following formula was applied to calculate the water content in the tissues: Water % = (W-D)/W X 100.

### Two-photon intravital microscopy (2P-IVM) of the brain and cervical spinal cord

**General surgery procedures for in vivo imaging.** Mice were administered a single dose of fentanyl (0.05 mg/kg)/midazolam (5 mg/kg)/medetomidine (0.5 mg/kg) via intramuscular injection in order to perform a tracheotomy and implant a carotid artery catheter (polyurethane 0.2 mm internal diameter, Ref BC-1P, Access Technologies, IL) as described before[24,35,76]. Arterial blood access allowed for systemic injection of exogenously labeled cells, thus allowing for immediate imaging[24,35,76]. During all surgical procedures and intravital imaging, vital parameters such as the heart rate (electrocardiogram (ECG)) and body temperature were constantly monitored and recorded. During the imaging period, we measured constant body temperatures around 36 °C and heart rates varying between individual mice from 250 beats per minute (bpm) as the lowest rate to 460 bpm as the highest rate measured. Mice were administered a single dose of buprenorphine (Temgesic; 0.3 mg/kg body weight) subcutaneously to provide adequate pain relief during surgical and imaging procedures and maintained under anesthesia via a tracheotomy using mechanical ventilation (Minivent, Model 845, Harvard Apparatus) with a gas mix of air and oxygen-containing 0.5–1% isoflurane at a tidal volume of 180 µl and 110 stokes/min. All 2P-IVM experiments of this study were terminal, and mice were euthanized directly after imaging by an overdose injection of ketamine (7.5 mg/mouse) and xylazine (0.5 mg/mouse) via the carotid catheter followed by decapitation and harvesting of organs of interest.

### Cervical spinal cord window and acute cranial window/skull thinning preparations

For all preparations, the surgery area was shaved and thoroughly disinfected. The cervical spinal cord window preparation allowing 2P-IVM

of the cervical spinal cord at the level of C3-C4 was performed as described in detail before[76]. After the preparation of the animals as described above, mice were placed in a prone position and fixed in a stereotactic frame. A skin incision was introduced from the neck to the bottom of the shoulder plate and muscle covering the spine was opened and fixed to the side of the stage using suture wire. The spine was cleaned of all muscles and a laminectomy was performed from C2 to C6 under a stereomicroscope. For brain imaging, we used both acute cranial window preparations and skull thinning as each preparation has its own limitations, as previously described by us[24] and others[77,78]. The acute cranial window and skull thinning preparations were performed as previously described[24]. For both cranial preparations, local analgesia was applied by administration of 6 mg/kg of lidocaine subcutaneously on top of the head 5 min prior skin opening. The skin was cut open over about one centimeter longitudinally between the ears and the periosteum was removed using surgical sponges (Questalpha Sugi sponge points Ref. 31603). A Microdrill (RWD Microdrill Model 780001 with 0.6 mm drill bits Part No 60-1000, CellPoint Scientific, Ideal microdrill burr set) was used to thin or open the skull between bregma and lambda on the right side of the sagittal suture of the skull over an area of 5 mm diameter for both types of cranial surgery. Overheating of the drilled area was prevented by cooling down the skull using cold saline flushing every 10 sec. For acute cranial window preparations, drilling was exclusively performed on the outer limit of the window until the bone was cut open. While the surgery area was superfused with 0.9% cold saline, the bone plate was carefully removed using a small sharp-edged spatula (AssutSuture), leaving the dura mater intact. The window was then protected and sealed with a round coverslip (diameter 6 mm #0, Epredia CB00060RA020MNZ0) and dental cement[79]. In contrast, for skull thinning, a circular region of the skull was homogenously drilled over the entire area until the bone thickness was below 50 µm. This surgery did not require additional protection of the brain as a portion of the skull was still covering it and thus maintaining a closed system.

### Two-photon intravital imaging (2P-IVM)

All imaging sessions were performed acutely right after surgery. 2P-IVM imaging was performed using a TrimScope-II two-photon laser scanning microscopy system (LaVision Biotec, Germany) equipped with an Olympus BX50WI fluorescence microscope and a water immersion objective (25X, NA 1.10, Nikon), equipped with a Spectra-Physics excitation laser with fixed 1045 nm and tunable 780-1300 nm laser lines, and the ImSpector software Pro64 v5.1.333 (LaVision Bio-Tec). Imaging area was set to 395 µm × 395 µm (512 pixels × 512 pixels) or 410 µm × 204 µm (1025 pixels × 512 pixels) XY, with imaging depth of 100-250 µm with 1-4 µm spacing corresponding to 25-250 z-frames every 30-130 s. Images were acquired at the different wavelengths of 860, 900, 920, 980, 1000, and 1045 nm or with 1045 nm in combination with one another wavelength to allow for adequate visualization of the used fluorophores. Aqp4-mRuby3 mice were imaged with an excitation wavelength of 1045 nm, the CNS border reporter mice were imaged simultaneously with excitation wavelengths of 920 and 1045 nm. Aqp4-mRuby3; CX3CR1-GFP mice were imaged with an excitation wavelength of 1000 nm. Finally, images of Aqp4-mRuby3 mice injected with anti-endoglin-AF633 antibody via carotid catheter and 10 kDa-FITC dextran via cisterna magna were obtained by overlaying images acquired with excitation wavelengths of 860 or 980 nm or simultaneous acquisition with 920 and 1045 nm when specified. To enable high quality 2P-IVM-imaging especially of the spinal cord, the acquisition of individual z-frames was synchronized with the induced mechanical ventilation of the mice, and tissue distortion and drift correction were performed using VivoFollow 2.0[80]. For tile imaging of several fields of view over time, two-photon multi-position imaging was used. In brief, four distinct fields of view were consecutively imaged per time point, and drift correction was performed for each tile

independently with VivoFollow. Image sequences were later aligned for each field of view during image post-processing. Sequences of image stacks were analyzed in the Fiji and the Imaris 9.8 software packages. Manual cell tracking of T cells was performed using Imaris v9.8 to v10.1 software; a minimum of 30 cells in each CNS compartment per mouse was considered for the analysis.

## Fluorescent beads, tracer or T cell infusion into the bloodstream or the cerebrospinal fluid compartment

During 2P-IVM imaging, the anesthetized, surgically prepared mice were systemically injected through the carotid artery with either an AF633 conjugated anti-endoglin antibody (40 μg/mouse), a 10 kDa AF680 dextran (20 μg/mouse, Thermo Fisher) or a 10 kDa FITC dextran (20 μg/mouse, Sigma-Aldrich, Switzerland) when specified to highlight the lumen of the blood vessels. $5 \times 10^6$ 2D2 Th1 were infused via the carotid artery catheter when specified. Where indicated, for tracer and bead infusion into the cisterna magna (CM), mice were implanted with a tracer and bead-filled cannula into the CM as previously described[24,81]. First, the dura mater was exposed by dissection of the neck muscles, allowing the insertion of the pre-filled cannula (polyurethane, Ref BC-1P, Access technologies, IL), mounted to a 30 G needle (Omnican 50, B.Braun) into the CM prior to surgery. The cannula was sealed to the dura and surrounding muscles (VetBond™, 3 M) prior to any further surgery. During 2P-IVM imaging, mice were infused with a total volume of 2.5 μl at a rate of 1 μL/min using an automated syringe pump (Harvard Apparatus, PHD ULTRA™ Syringe Pump) with a solution of 2.5 μg 10 kDa FITC dextran (Invitrogen D1820) and $9 \times 10^6$ PEGylated beads as previously described[82].

## Statistics and reproducibility

GraphPad Prism v9.2 and v10.3.1 software (La Jolla, CA, USA) was used to perform statistical analysis. The respective statistical tests used are specified in each figure legend. Data are presented as mean ± SD or ± SEM and precise p values are indicated where statistical significance was reached.

Quantifications in Fig. 2b–d are from a total of 3 independent experiments. The number of independent experiments on which quantifications in Figs. 2h, 9g, h, 10h–k are based are indicated in the figure legends. Quantifications in Fig. 3b, c, Fig. 2e and Supplementary Fig. 3 are based on 2 independent experiments.

## Reporting summary

Further information on research design is available in the Nature Portfolio Reporting Summary linked to this article.

# Data availability

All data are available in the main text or supplementary materials. Imaging datasets will be made available upon request. Source data are provided with this paper.

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

## Acknowledgements

This study was funded by the Fidelity Bermuda Foundation and the Swiss National Science Foundation (grant N° 310030_189080 and CRSII5_213535 to BE; grant N° 310030_189226 to STP) and by a Seal of Excellence Fund (SELF) of the University of Bern to PHL and BE. We would like to specifically acknowledge the late Roy Weller (University of Southampton, UK) for his valuable advice during this study. Additional thank goes to Christel Genoud (University of Lausanne, Switzerland) for her valuable advice on the analysis of astrocyte endfeet ultrastructure. We also thank Simon Aleandri and Paola Luciani (University of Bern, Switzerland) for the PEGylation of the microbeads. We thank Doron Merkler and Nicolas Page (University of Geneva, Switzerland) for providing LCMV-OVA. We acknowledge the Microscopy Imaging Center of the University of Bern and specifically Marek Kaminek (Institute of Anatomy, University of Bern) for support in performing the transmission electron microscopy studies. We furthermore express our sincere thanks to the dedicated work of the animal caretakers for professionally maintaining the transgenic mouse colonies.

## Author contributions

Conceptualization: B.E., U.D., M.V., J.P., P.H.L., B.Z. Methodology: P.H.L., E.B., M.V., J.P., F.K., J.M., A.O., Y.O., I.S., S.P., S.B., A.mB., A.l.B., B.H., U.D., B.Z. Investigation: P.H.L., F.K., J.M., E.B., A.O., M.V., J.P., Y.O., I.S., S.B., B.H., U.D., B.Z. Visualization: P.H.L., F.K., J.P., E.B., A.O., M.V. Funding acquisition: B.E., P.H.L., S.T.P. Supervision: B.E., U.D., S.T.P., B.Z. Writing of the manuscript: P.H.L., F.K., J.P., E.B., Y.O., C.F., B.H., S.T.P., B.E., U.D.

## Competing interests

The authors declare no competing interests.
