## [Transparent Peer Review file · Nature Communications]

***In vivo* imaging of the barrier properties of the glia limitans during health and neuroinflammation**

Corresponding Author: Professor Britta Engelhardt

Version 0:

Reviewer comments:

Reviewer #1

(Remarks to the Author)

In this paper, the authors introduce the Aqp4-mRuby3 reporter mouse, which allows for ex vivo and in vivo visualization of the glia limitans in brain and spinal cord. By combining the Aqp4-mRuby3 mouse with advanced imaging techniques such as 2-photon intravital microscopy (2P-IVM) and crossing it with other reporter lines (e.g., VE-cadherin-GFP), the authors create a CNS border reporter mouse capable of simultaneous imaging of what the authors refer to as the glia limitans. They find that heterozygous expression of the AQP4-mRuby3 fusion protein does not affect CNS water balance, astrocyte end-foot morphology, or EAE pathogenesis. They also use the combination of this reporter mouse with other lines to explore the interaction of the glia limitans with the meninges. Furthermore, they demonstrate the glia limitans' barrier properties, preventing tracer and immune cell penetration into the CNS parenchyma under steady-state conditions, while showing that antigen recognition allows CD8 T cells to cross this barrier during neuroinflammation. This study highlights the utility of the Aqp4-mRuby3 mouse in advancing our understanding of CNS border dynamics and immune surveillance.

The validation of this mouse is thorough but the initial application describing immune cell infiltration across the GL is not convincing. I also have conceptual questions.

Comments:

1. I wonder what the authors define as the glia limitans. Given that the superficial GL is cell bodies, and the capillary GL is processes, and both create a basement membrane, what structures exactly represent the GL? Indeed the authors call the broad band of Aqp4 signal at the brain surface the superficial GL but this signal could be L1 astrocytes, processes of superficial astrocytes or a combination of both. This should be clarified in the MS.
2. In figure 4B (second panel to the right) they show Dextran staining in the form of processes, but this structure which presumably is the meninges (?), is not described. It would be useful for the reader to know what these structures are.
3. In figures 4B and C (first panel) they show diffuse Aqp4 expression at the brain surface. Do the authors think Aqp4 is less polarized in the membrane of the superficial GL cell bodies? What would this mean in terms of the function of the superficial GL? Are they involved in the glymphatic system?
4. Do the authors have a way to visualise the transition from surface GL to perivascular GL using this mouse (something similar to Fig10i)?
5. In several figures expression of Aqp4 is seen around the subpial blood vessels, but the possible role of the glia limitans in these structures is not mentioned. Is this a type of glia limitans at superficial blood vessels?
6. The figures describing the presence of T cells at brain borders vs in the parenchyma are not clear. Is there a way to quantify the infiltration of immune cells across the GL in a meaningful way? Only showing a single T cell in the parenchyma in their neuroinflammation model is not sufficient.

Typos:

Line 118:

Original: "heterozygous Aqp4-mRuby mice"

Correction: "heterozygous Aqp4-mRuby3 mice" (typo in "mRuby3").

Line 280:

Original: "ameboid-like"

Correction: "amoeboid-like" (correct spelling).

Reviewer #2

(Remarks to the Author)

In the manuscript entitled 'In vivo imaging of the barrier properties of glia limitans during health and neuroinflammation' by H elie-Legoupil et al has put forward an in depth characterization of a new mouse line, Aqp4-mRuby3 knock-in mice, to study the glia limitans and perivascular borders created by astrocytes. The creation of this mouse line allowed for more adequate observations of the glia limitans and perivascular borders using in vivo imaging approaches. Further, they were able to cross these mice with a number of GFP reporter lines, VE-cadherin-GFP & Cx3cr1-GFP mouse strains. This opens the possibility to simultaneously visualize barriers created by astrocytes, arachnoid barriers, blood-brain barriers, as well as the compartmentalization of different immune cells like border associated macrophages, microglia and T cells. Indeed, the authors demonstrated that the border mice (Aqp4-mRuby3; VE-cad-GFP) were particularly useful for studying the dynamics and migratory behaviors of T cells in neuroinflammatory mouse models within and through the different barrier compartments. The studies help demonstrate how these mice could grow in their usefulness to study a multitude of neuroinflammatory conditions. I have a few comments to help the authors strengthen these studies outlined below:

1. The authors provide nice electron microscopy images and quantification of astrocytic endfeet swelling surrounding vessels in Aqp4-mRuby3/+ and +/- mice and they show the swelling is comparable. However, is this the case along the glia limitans? Is astrocyte cell body and endfeet morphology similar there in the heterozygous mice? Sometimes perfusion/fixation efficiency can influence the swelling of astrocytic endfeet and so it is possible that swelling due to this happened in some wildtype mice and that masks the affect in the heterozygous mice. So, does the scoring of endfeet swelling look comparable across the animals within each genotype group? As of now the scores of the individual vessels are reported.
2. In Fig 2h. How do you know those are perivascular CD45+ cells? That area is quite large, and it is difficult to see where the vessel is/was. Could it just be a lesion site?
3. For the i.c.m. injections of the tracer and beads, the animals were under isoflurane-induced anesthesia during these experiments according to the methods. This can likely affect how substances can move within the SAS and also attenuates vasomotor oscillations. So, while these experiments were important for showing the glia limitans function in likely prohibiting the movement of tracer and beads, the dynamics shown in Fig. 8h should be taken with the caveat that this was under anesthesia conditions. Further without having appropriate vasomotor oscillations due to the use of anesthesia, it is difficult to state whether the tracer or beads were limited in their movement and unable to enter the periarterial space due to this. The statement on lines 295-296 page 9 could be misconstrued that the tracer did not enter the periarterial space due to astrocytic borders: "The Aqp4-mRuby3 signal highlighted that the 10kDa FITC dextran remained restricted to the SAS above the superficial glia limitans and did not enter the penetrating periarterial space." A caveat to these experiments should be provided and it should be stated in the results that animals for i.c.m. injections and tracer dynamic studies were kept under anesthesia during these experiments. The barrier properties are likely the same, but the physiology of tracers could be different, just trying to maintain clarity for comparison reasons across many studies that look at CSF dynamics.
4. The authors describe the location of CD8 T cells within the dura, within the border compartment, perivascular spaces, and parenchyma. But there is no quantification to support these observations in Fig. 10. Could the authors provide quantification to understand the proportionality of the CD8 T cell trafficking in their model of autoimmune neuroinflammation (ODC-OVA mice subjected to peripheral infection with LCMV-OVA to activate OVA-specific T cells). The authors describe this mouse model on lines 330-332 of page 10 as eventually OVA-specific T cells entering the brain parenchyma. However, it is interesting that the authors note that very few T cells have entered the parenchyma by day 7 but this was not the case in their previous related work (PMID: 37730744).
5. Similarly, the authors state that CD8 T cell motility is reduced at day 7 but no quantification is provided to indicate this is the case day 1 vs day 7. Quantification of this would greatly support these statements. Additionally, in Supplementary Movie 12, it is clear that some CD8 T cells that are focused on appear to not move in the XYZ but there still are some that are traveling greatly within the XY plane. Is there a difference in T cell dynamics when they are traversing in the Z plane (through the AB or glia limitans) vs. the XY planes?
6. On lines 482-483 page 14, the authors state that their data supports a model whereby the glia limitans is created by astrocyte cell bodies. I'm not sure where in these studies this was shown? Could a deeper exploration in the electron microscopy images help with this?
7. Figures 9 & 10 could be a little easier to follow if cartoons of their neuroinflammatory models, their paradigms, and time points were provided. And if the timepoints could be added alongside the images that would greatly help the reader follow.

Minor edits

1. OAPs are defined twice in the manuscript.
2. Perivascular is misspelled on line 480 of page 14.

Reviewer #3

(Remarks to the Author)

In this work, Hélie-Legoupil et al develop and characterize a mouse model expressing AQP4-mRuby that allows for intravital imaging of the glia limitans during health and disease. The authors first perform a careful characterization of the mice to ascertain the expected localization of the fluorescent signal and confirm that at least in heterozygote mice both the water balance as well as the ultrastructure of astrocyte end feet are not altered. Next, the authors perform intravital imaging both of the brain and spinal cord to show that the reporter expression outlines the localization of the glial limitans in vivo. Finally, the authors generate a "CNS border reporter mouse" by crossing the AQP4-mRuby knock-in mice (which labels the glia limitans) with a previously established VE-cadherin GFP reporter line (which labels the vascular, pial and arachnoid CNS borders). Using this CNS border reporter mouse the authors then show that they can differentiate CNS border patrolling from CNS infiltrating T cells in models of CD4+ and CD8+ T cell-driven neuroinflammation. Since T cell infiltration to the CNS is a critical step in neuroinflammatory conditions and beyond, tools that allow a more refined characterization of the process in vivo should be quite useful to the community. Overall this is thus a very straightforward paper that establishes a useful new tool that provides an important advantage, namely the ability to assign the localization of immune cells to intravascular, perivascular and parenchymal positions in vivo. Further strengths of the paper are the extensive characterization of this model using both cortical and spinal imaging approaches and both models driven by CD4+ and CD8+ T cells. I have only the following comments:

1) While I understand that a main application of this tool is to determine the localization of cells with regards to the glia limitans, it would of course also be helpful if changes to the structure of the glial limitans e.g. during neuroinflammation could be tracked more directly (e.g to quantify this in vivo or relate it to the pattern of immune cell infiltration). Can this be achieved ?

2) As the main strength of the approach is to localize infiltrating immune cells it would be good to improve on the in vivo images shown in Figure 9 a-c. While the T cell localization is well shown in panel D (which however is obtained by confocal imaging of vibratome sections that does not require an in vivo label), the outline of the glia limitans is not clearly discernable for me in the example images currently provided. This should be feasible as the glial limitans can be identified much clearer in the example images provided in Figure 10.

3) In Figure 3, the authors make the point that the end feet of the astrocytes with the AQP4-mRuby are unaffected compared to WT astrocytes. They use a swelling score and they compare using the mean and standard deviation between KI and WT mice. Since it's a scale with qualitative values, it would be better to show proportion of end feet of each type per mouse per genotype, as the numbers are just coding qualitative characteristics.

Reviewer #4

(Remarks to the Author)

Reviewer #5

(Remarks to the Author)

Version 1:

Reviewer comments:

Reviewer #1

(Remarks to the Author)

I thank the authors for addressing all my concerns. This will be an incredibly valuable tool for the astrocyte community and beyond.

Reviewer #2

(Remarks to the Author)

The resubmission of 'In vivo imaging of the barrier properties of glia limitans during health and neuroinflammation' by Hélie-Legoupil, Kloster et al, I believe, has improved upon their overall characterization of the Aqp4-mRuby3 knock-in mice to

study barrier compartments within the CNS. Especially regarding the detailed analysis of T cells within the various barrier compartments and their dynamics during immune surveillance and neuroinflammation. The addition of serial block face electron microscopy to show the composition of the superficial glia limitans does highlight the complexity of astrocytes creating that tissue as well. I do have a few lingering thoughts on our discussion.

1. The authors adequately addressed the possibility for differences between astrocytic endfeet swelling among individual WT and het mice for the perivascular glia limitans and demonstrated that potentially poor tissue fixation in WTs was unlikely to mask a swelling effect in the het mice. Further, the authors nicely add volumetric electron microscopy (EM) and show qualitatively that astrocytes and their processes are likely unaffected in the het mice along the superficial glia limitans. I realize these were performed on one animal per genotype given the constraints with volume EM, as well as the necessity of this approach due to the complexity of the glia limitans there. Is it possible to provide some sort of quantification to accompany the conclusions made on the volume EM here? Perhaps measuring the volume of individual cell bodies and processes among the two animals?

2. The addition of laminin to the images in Fig. 2g helps greatly to demonstrate the perivascular cuffing occurring in EAE. Do you see similar amounts of perivascular cuffing to this magnitude in WT vs. Aqp4-mRuby het mice with EAE? This would bolster the conclusion that heterozygous Aqp4-mRuby mice have comparable neuroinflammatory responses.

3. For the possible complications of anesthesia altering CSF flow, I appreciate the discussion regarding the low-dose isoflurane potentially having minimal effect on CSF flow. I missed the part where the animals were also under fentanyl/midazolam/medetomidine mixture as well. We use the exact same dosage in our animals for surgeries as well and it very much slows down breathing and heart rate, both of which are known drivers of CSF mobility throughout the brain. Unless the authors have heart rate and breathing metrics to accompany these experiments indicating these physiological processes were functioning similarly to an awake mouse during the 2P imaging, I think it would be best if the authors omit their conclusions in challenging the entry of CSF into the periarterial spaces here and add the details of the F/M/M cocktail on heart and breathing physiology. There are too many parameters that are altered in these animals in these experiments to conclude that the 10kDa FITC and/or beads would not have ever entered the periarterial spaces and thus challenging the entry of CSF into the periarterial spaces.

Reviewer #3

(Remarks to the Author)

The authors have further improved their study and all my concerns have been well addressed. I support the publication of this work.

Reviewer #4

(Remarks to the Author)

Reviewer #5

(Remarks to the Author)

Author's Response to Reviewers

Reviewer #1

In this paper, the authors introduce the Aqp4-mRuby3 reporter mouse, which allows for ex vivo and in vivo visualization of the glia limitans in brain and spinal cord. By combining the Aqp4-mRuby3 mouse with advanced imaging techniques such as 2-photon intravital microscopy (2P-IVM) and crossing it with other reporter lines (e.g., VE-cadherin-GFP), the authors create a CNS border reporter mouse capable of simultaneous imaging of what the authors refer to as the glia limitans. They find that heterozygous expression of the AQP4-mRuby3 fusion protein does not affect CNS water balance, astrocyte end-foot morphology, or EAE pathogenesis. They also use the combination of this reporter mouse with other lines to explore the interaction of the glia limitans with the meninges. Furthermore, they demonstrate the glia limitans' barrier properties, preventing tracer and immune cell penetration into the CNS parenchyma under steady-state conditions, while showing that antigen recognition allows CD8 T cells to cross this barrier during neuroinflammation. This study highlights the utility of the Aqp4-mRuby3 mouse in advancing our understanding of CNS border dynamics and immune surveillance.

The validation of this mouse is thorough but the initial application describing immune cell infiltration across the GL is not convincing. I also have conceptual questions.

Comments:

1. I wonder what the authors define as the glia limitans. Given that the superficial GL is cell bodies, and the capillary GL is processes, and both create a basement membrane, what structures exactly represent the GL? Indeed the authors call the broad band of Aqp4 signal at the brain surface the superficial GL but this signal could be L1 astrocytes, processes of superficial astrocytes or a combination of both. This should be clarified in the MS.

Answer: We define the glia limitans as the outermost layer of tissue of brain and spinal cord lying directly below the pia mater and thus forming the border between the non-neuronal tissue and the CNS parenchyma. At the cellular level the glia limitans is composed of the parenchymal basement membrane and the astrocytes localized at this border. It is well accepted that at the perivascular level astrocyte end-feet project to this tissue border and this is confirmed by the ultrastructural data shown in our present study (Figure 3).

In contrast, the superficial glia limitans of mouse brain and spinal cord was previously shown to rather be comprised a layer of surface astrocytes (DOI: 10.1002/ar.22717) and a recent study by the laboratory of Shane Liddelow confirmed this observation and identified a specialized astrocyte subtype expressing myocilin at the superficial glia limitans (DOI: 10.1016/j.celrep.2025.115344). We have clarified this definition in the revised manuscript and have added additional ultrastructural and immunofluorescence data (Supplementary Figures 4 and 5) underscoring that at the superficial glia limitans we find astrocyte cell bodies extending long processes along this surface. Furthermore, we now show that the AQP4-mRuby3 signal on the superficial glia limitans astrocytes is polarized towards the surface (Supplementary Figure 5).

2. In figure 4B (second panel to the right) they show Dextran staining in the form of processes, but this structure which presumably is the meninges (?), is not described. It would be useful for the reader to know what these structures are.

Answer: The dextran signal shows the lumen of the blood vessels. The "processes" the Reviewer refers to is the second harmonic generation signal (SHG) of the collagen type I fibers in the dura mater that when imaging with an excitation wavelength of 1045 nm as done here is visible in green. This is explained in the Results and we have edited the Figure legend to make this clearer.

3. In figures 4B and C (first panel) they show diffuse Aqp4 expression at the brain surface. Do the authors think Aqp4 is less polarized in the membrane of the superficial GL cell bodies? What would this mean in terms of the function of the superficial GL? Are they involved in the glymphatic system?

4. Do the authors have a way to visualise the transition from surface GL to perivascular GL using this mouse (something similar to Fig10i)?

Answer: We thank the Reviewer for these excellent points 3 and 4 which we have addressed with additional experiments and answer here together. To address both issues, we have performed additional

immunofluorescence stainings on decalcified coronal skull cryosections from heterozygous Aqp4-mRuby3 reporter mice and analyzed those by confocal microscopy. These data were added as Supplementary Figure 5 and show polarized localization of AQP4-mRuby3 towards the outer surface of the superficial glia limitans in two different regions of the brain. At the level of the perivascular glia limitans the AQP4-mRuby3 signal is clearly polarized to the astrocyte end-feet around the blood vessels (as depicted in Supplementary Figure 2). At the surface of the brain the AQP4-mRuby3 signal is polarized towards the outer surface on the astrocyte cell bodies forming the superficial glia limitans (Supplementary Figure 5). The reporter mouse thus also recapitulates the AQP4 immunostaining as observed by immunofluorescence staining of the brain in WT healthy mice (Supplementary Fig 2a). Furthermore, we have added 2P-IVM imaging data in Supplementary Figure 5c highlighting the transition zone from the superficial to the perivascular glia limitans which confirms the continuous AQP4-mRuby3 signal towards the outer surface of the CNS parenchyma.

Please note that in this context we have also addressed the composition of the superficial glia limitans at the ultrastructural level and confirm previous reports that this is rather formed by astrocyte cell bodies localized at the surface extending large processes (Supplementary Figure 4). These explanations have been added to the Results section.

Last but not least, please note that we cannot confirm the concept of the glymphatic hypothesis where fluorescent tracers enter from the subarachnoid space periaxonal spaces in the brain – see Figure 8 and Supplementary Figure 7.

5. In several figures expression of Aqp4 is seen around the subpial blood vessels, but the possible role of the glia limitans in these structures is not mentioned. Is this a type of glia limitans at superficial blood vessels?

Answer: We apologize to this Reviewer as it is not entirely clear what is referred to here. In Figure 4a the 2P-IVM shows dextran filled blood vessels above the superficial glia limitans but then e.g. at the left side on vessel diving into the parenchyma where a sharp ring of the AQP4-mRuby3 signal is visible – this would be the transition zone of the superficial to the perivascular glia limitans addressed above. In Figure 5 the AQP4-mRuby3 signal from the superficial glia limitans is seen to embrace the inner part of the blood vessel only while the blood vessels in the parenchyma are entirely surrounded by the AQP4-mRuby3 signal of the perivascular glia limitans. Supplementary Figure 5 in the revised manuscript now shows a more detailed analysis of the transition zone of the superficial to the perivascular glia limitans where we found that the AQP4-mRuby3 signal remains polarized towards the outer surface of the CNS parenchyma and is thus continuous. We hope these additional data and our comments clarify the issue.

6. The figures describing the presence of T cells at brain borders vs in the parenchyma are not clear. Is there a way to quantify the infiltration of immune cells across the GL in a meaningful way? Only showing a single T cell in the parenchyma in their neuroinflammation model is not sufficient.

Answer: We have added quantification of GFP⁺ 2D2 CD4 T cells during experimental autoimmune encephalomyelitis (EAE) in the different CNS compartments that can be identified in the CNS border reporter mice by 2P-IVM, namely the parenchyma, within blood vessels, the subpial and the subarachnoid space and the dura mater (Figure 9g) and analyzed their dynamic behaviour (Figure 9h). In Figure 9i additional representative confocal images of those mice following 2P-IVM are provided to show the precise localization of the CD4 T cells relative to these barrier and border landmarks ex vivo.

Moreover, quantification of GFP⁺ OT-I CD8 T cells during 2P-IVM in the Aqp4-mRuby3 reporter with respect to the glia limitans (this is above: in border compartments vs. below: in parenchyma) has been performed. Here we compared the conditions of CNS immune surveillance and neuroinflammation in the CD8 T cell mediated-autoimmune neuroinflammation model. These data have been added as Figure 10h.

Typos:

Line 118:

Original: "heterozygous Aqp4-mRuby mice"

Correction: "heterozygous Aqp4-mRuby3 mice" (typo in "mRuby3").

Line 280:

Original: "ameboid-like"

Correction: "amoeboid-like" (correct spelling).

Answer: We apologize for these typing errors which have been corrected in the revised version of the manuscript.

Reviewer #2:

In the manuscript entitled 'In vivo imaging of the barrier properties of glia limitans during health and neuroinflammation' by Hélie-Legoupil et al has put forward an in depth characterization of a new mouse line, Aqp4-mRuby3 knock-in mice, to study the glia limitans and perivascular borders created by astrocytes. The creation of this mouse line allowed for more adequate observations of the glia limitans and perivascular borders using in vivo imaging approaches. Further, they were able to cross these mice with a number of GFP reporter lines, VE-cadherin-GFP & Cx3cr1-GFP mouse strains. This opens the possibility to simultaneously visualize barriers created by astrocytes, arachnoid barriers, blood-brain barriers, as well as the compartmentalization of different immune cells like border associated macrophages, microglia and T cells. Indeed, the authors demonstrated that the border mice (Aqp4-mRuby3; VE-cad-GFP) were particularly useful for studying the dynamics and migratory behaviors of T cells in neuroinflammatory mouse models within and through the different barrier compartments. The studies help demonstrate how these mice could grow in their usefulness to study a multitude of neuroinflammatory conditions. I have a few comments to help the authors strengthen these studies outlined below:

Answer: We thank this Reviewer for the positive remark and the intention to help us improve our manuscript with the critiques. We have addressed the points as outlined below.

1. The authors provide nice electron microscopy images and quantification of astrocytic endfeet swelling surrounding vessels in Aqp4-mRuby3/+ and +/+ mice and they show the swelling is comparable. However, is this the case along the glia limitans? Is astrocyte cell body and endfeet morphology similar there in the heterozygous mice? Sometimes perfusion/fixation efficiency can influence the swelling of astrocytic endfeet and so it is possible that swelling due to this happened in some wildtype mice and that masks the affect in the heterozygous mice. So, does the scoring of endfeet swelling look comparable across the animals within each genotype group? As of now the scores of the individual vessels are reported.

Answer: The perivascular astrocyte end-feet are part of the perivascular glia limitans – thus astrocyte end-feet morphology in the perivascular glia limitans is comparable as shown.

Perfusion and fixation does indeed have an impact on astrocyte end-foot morphology. The blinded analysis of the astrocyte end-foot swelling at the ultrastructural level identified one mouse where numerous astrocyte endfeet were swollen, which correlated with incomplete perfusion as determined by the detection of erythrocytes in numerous capillaries. This is why this mouse was excluded and we here show the valid data from 2 heterozygous Aqp4-mRuby3 mice and 3 WT mice.

By showing the distribution of the swelling score of astrocyte end-feet across genotypes for each individual mouse in different regions of the brain and analyzing in a blinded fashion we are confident that the data shown are not affected by sample preparation artifacts (Figure 3c). The distribution of the swelling scores is indeed comparable between heterozygous Aqp4-mRuby3 and WT mice.

We have furthermore revised Figure 3 allowing to see the swelling score of astrocyte endfeet from each individual mouse (Figure 3c).

2. In Fig 2h. How do you know those are perivascular CD45+ cells? That area is quite large, and it is difficult to see where the vessel is/was. Could it just be a lesion site?

Answer: We have revised Figure 2 to improve clarity. Perivascular cuffing of infiltrating immune cells is a characteristic hallmark of EAE. The revised Figure 2g now shows in addition to the AQP4-mRuby3 signal and immunostaining for CD45, immunostaining for laminin allowing for visualization of the endothelial and parenchymal basement membranes bordering the perivascular space. The parenchymal basement membrane is part of the glia limitans. The additional laminin staining confirms perivascular localization of the majority of CD45+ immune cells.

3. For the i.c.m. injections of the tracer and beads, the animals were under isoflurane-induced anesthesia during these experiments according to the methods. This can likely affect how substances can move within the SAS and also attenuates vasomotor oscillations. So, while these experiments were important for showing the glia limitans function in likely prohibiting the movement of tracer and beads, the dynamics shown in Fig. 8h should be taken with the caveat that this was under anesthesia conditions. Further without having appropriate vasomotor oscillations due to the use of anesthesia, it is difficult to state whether the tracer or beads were limited in their movement and unable to enter the periarterial space due to this. The statement on lines 295-296 page 9 could be misconstrued that the tracer did not enter the periarterial space due to astrocytic borders: "The Aqp4-mRuby3 signal highlighted that the 10kDa FITC dextran remained restricted to the SAS above the

superficial glia limitans and did not enter the penetrating periarterial space.” A caveat to these experiments should be provided and it should be stated in the results that animals for i.c.m. injections and tracer dynamic studies were kept under anesthesia during these experiments. The barrier properties are likely the same, but the physiology of tracers could be different, just trying to maintain clarity for comparison reasons across many studies that look at CSF dynamics.

Answer: *Our experiments were performed under low-dose isoflurane (0.5% to 1%), after an initial injection anesthesia cocktail (mixture of fentanyl (0.05 mg/kg)/midazolam (5 mg/kg)/medetomidine (0.5 mg/kg)). Indeed, we do not know exactly how this anesthesia protocol affects the CSF distribution of the tracers. However, to this end inhibition of CSF flow has only been observed with higher doses of isoflurane (2-3%). This was originally reported by the Benveniste group using MRI (DOI: 10.1097/ALN.0000000000001888) and later confirmed by Steven Proulx – a coauthor on this manuscript (DOI: 10.1007/s00401-018-1916-x). In a recent MRI study, a higher dose of isoflurane (2-3%) inhibited efflux from the ventricles compared to lower dose (1-1.5%) (DOI: 10.1002/adv.202501502). The study of the Proulx lab directly comparing high and low doses of isoflurane anesthesia showed that high isoflurane doses (2-3%) also prohibited cisterna magna injected tracer movement into the cortical SAS. As we did observe tracer movement in this space we can at least exclude that the anesthesia protocol used in the present study limits the previously reported detrimental effects on CSF flow with this anesthetic. We have added mentioning of this limitation of our study to the Discussion.*

4. The authors describe the location of CD8 T cells within the dura, within the border compartment, perivascular spaces, and parenchyma. But there is no quantification to support these observations in Fig. 10. Could the authors provide quantification to understand the proportionality of the CD8 T cell trafficking in their model of autoimmune neuroinflammation (ODC-OVA mice subjected to peripheral infection with LCMV-OVA to activate OVA-specific T cells). The authors describe this mouse model on lines 330-332 of page 10 as eventually OVA-specific T cells entering the brain parenchyma. However, it is interesting that the authors note that very few T cells have entered the parenchyma by day 7 but this was not the case in their previous related work (PMID: 37730744).

Answer: *We have added quantification of GFP⁺ OT-I CD8 T cells during 2P-IVM in the Aqp4-mRuby3 reporter with respect to the glia limitans (this is above: in border compartments vs. below: in parenchyma). We specifically compared the conditions of CNS immune surveillance and neuroinflammation in the CD8 T cell mediated-autoimmune neuroinflammation model. These data have been added as Figure 10h.*

In addition, we have quantified GFP⁺ 2D2 CD4 T cells during experimental autoimmune encephalomyelitis (EAE) in the different CNS compartments that can be identified in the CNS border reporter mice by 2P-IVM, namely the parenchyma, within blood vessels, the subpial and the subarachnoid space and the dura mater (Figure 9g) and analyzed their dynamic behaviour (Figure 9h). In Figure 9i additional representative confocal images of those mice following 2P-IVM are provided to show the precise localization of the CD4 T cells relative to these barrier and border landmarks ex vivo.

With respect to the second query where the Reviewer addresses an apparent discrepancy between our present and previously published observation we would like to point out that in the previous study – Mapunda, Pareja et al PMID: 37730744 – quantification of CD8 T cells was performed in VE-cadherin-GFP mice which do not allow to distinguish between the subpial and the parenchymal space. Please note that therefore, the CD8 T cells quantified by 2P-IVM in the subpial space in the previous study are not necessarily in the parenchyma, since they could still be localized in the subpial space, above the glia limitans. In this previous study we did indeed see few subpial cells under the conditions of CNS immune surveillance. This is in accordance with our observations in the present study where we did observe a very low number of T cells (<10/FOV) at the glia limitans and below under conditions of CNS immune surveillance. These are very low numbers when compared to conditions of neuroinflammation where we found >300 cells/FOV at and below the glia limitans.

5. Similarly, the authors state that CD8 T cell motility is reduced at day 7 but no quantification is provided to indicate this is the case day 1 vs day 7. Quantification of this would greatly support these statements.

Additionally, in Supplementary Movie 12, it is clear that some CD8 T cells that are focused on appear to not move in the XYZ but there still are some that are traveling greatly within the XY plane. Is there a difference in T cell dynamics when they are traversing in the Z plane (through the AB or glia limitans) vs. the XY planes?

Answer: *With respect to the first point, we apologize to this Reviewer as it seems that our description was not clear. We rather meant that at day 7 p.i. CD8 T cell motility is reduced in the condition of neuroinflammation*

when compared to the condition of CNS immune surveillance. Adding additional quantification of CD8 T cell tracking to the revised manuscript further underscores this observation and is now shown in Figure 10i. In fact, CD8 T cell movement above and below the glia limitans is reduced under conditions of neuroinflammation when compared to CNS immune surveillance at the same day pi.

With respect to the second point, we can say that indeed the T cells were observed to be more motile in XY, therefore the displacement of CD8 T cells measured and provided now in Fig. 10i is mainly of the XY plane of imaging. We did not observe any CD8 T cells crawling on top of the glia limitans and suddenly detaching or attaching/detaching suggesting that they are indeed in the subpial space and have no longer direct access to the CSF filled subarachnoid space. T cells below the glia limitans were less restricted to crawling in the XY plane and showed more prominent Z displacement. The revised manuscript now includes quantification of T cell crawling events in the different compartments (Figure 9 and 10).

We also observed T cell migration in the Z-plane – see Supplementary Movie 15 which shows T cell migration across the glia limitans. However these events were too rare and thus prohibited a meaningful quantification.

6. On lines 482-483 page 14, the authors state that their data supports a model whereby the glia limitans is created by astrocyte cell bodies. I'm not sure where in these studies this was shown? Could a deeper exploration in the electron microscopy images help with this?

Answer: In contrast to the perivascular glia limitans which is formed by astrocyte-end feet and the parenchymal basement membrane, the superficial glia limitans of mouse brain and spinal cord was previously shown to rather be comprised a layer of surface astrocytes (DOI: 10.1002/ar.22717) and a recent study by the laboratory of Shane Liddelow confirmed this observation and identified a specialized astrocyte subtype expressing myocilin at the superficial glia limitans (DOI: 10.1016/j.celrep.2025.115344). We have better highlighted this information in our revised manuscript and have added additional ultrastructural and immunofluorescence data (Supplementary Figures 4 and 5) underscoring that at the superficial glia limitans we find astrocyte cell bodies extending long processes along this surface. Additionally, a novel Supplementary Movie 3 now shows the complete z-stack of the superficial glia limitans of a heterozygous Aqp4-mRuby3 mouse and allows to see the astrocyte cell bodies and their cellular processes forming the superficial glia limitans.

7. Figures 9 & 10 could be a little easier to follow if cartoons of their neuroinflammatory models, their paradigms, and time points were provided. And if the timepoints could be added alongside the images that would greatly help the reader follow.

Answer: We have added schemes for each experimental setup of the respective neuroinflammation models used and the disease timepoints analyzed to Figures 2, 9 and 10.

Minor edits

1. OAPs are defined twice in the manuscript.

Answer: Definition of OAPs is now limited to the discussion.

2. Perivascular is misspelled on line 480 of page 14.

Answer: We thank this Reviewer for pointing out these errors which have been corrected.

Reviewer #3:

In this work, Hélié-Légoupil et al develop and characterize a mouse model expressing AQP4-mRuby that allows for intravital imaging of the glia limitans during health and disease. The authors first perform a careful characterization of the mice to ascertain the expected localization of the fluorescent signal and confirm that at least in heterozygote mice both the water balance as well as the ultrastructure of astrocyte end feet are not altered. Next, the authors perform intravital imaging both of the brain and spinal cord to show that the reporter expression outlines the localization of the glial limitans in vivo. Finally, the authors generate a “CNS border reporter mouse” by crossing the AQP4-mRuby knock-in mice (which labels the glia limitans) with a previously established VE-cadherin GFP reporter line (which labels the vascular, pial and arachnoid CNS borders). Using this CNS border reporter mouse the authors then show that they can differentiate CNS border patrolling from CNS infiltrating T cells in models of CD4+ and CD8+ T cell-driven neuroinflammation. Since T cell infiltration to the CNS is a critical step in neuroinflammatory conditions and beyond, tools that allow a more refined characterization of the process in vivo should be quite useful to the community. Overall this is thus a very straightforward paper that establishes a useful new tool that provides an important advantage, namely the

ability to assign the localization of immune cells to intravascular, perivascular and parenchymal positions in vivo. Further strengths of the paper are the extensive characterization of this model using both cortical and spinal imaging approaches and both models driven by CD4+ and CD8+ T cells.

Answer: We thank this Reviewer for the overall positive feed-back on our study.

I have only the following comments:

1) While I understand that a main application of this tool is to determine the localization of cells with regards to the glia limitans, it would of course also be helpful if changes to the structure of the glial limitans e.g. during neuroinflammation could be tracked more directly (e.g to quantify this in vivo or relate it to the pattern of immune cell infiltration). Can this be achieved ?

Answer: We thank the Reviewer for this excellent thought. We have assessed the options for quantification of structural changes of the glia limitans based on the AQP4-mRuby3 signal by 2P-IVM. We found that the limited resolution of 2P-IVM – especially in the Z-axis – makes this a very difficult task to begin with. During neuroinflammation additional issues with respect to appearance of unspecific autofluorescent signals further impair such an analysis. Please also note that the AQP4-mRuby3 signal at the superficial glia limitans is quite fuzzy to begin with which complicates efforts for quantification. To this end we were not able to detect significant changes of the superficial AQP4-mRuby3 signal by 2P-IVM, however, this may be possible in a novel study aiming to determine this possibility by exploring additional algorithms for signal quantification over time. However, making use of the border reporter mouse expressing VE-cadherin-GFP and AQP4-mRuby3 we were able to image the enlargement of perivascular spaces, delineated by the endothelial VE-cadherin-GFP signal and AQP4-mRuby3 signal from the glia limitans in neuroinflammation (peak aEAE) as now shown in Supplementary Movie 12. Also, please note the Aqp4-mRuby3 reporter mouse truthfully reproduced the previously reported loss of perivascular AQP4 under conditions of neuroinflammation as shown by confocal microscopy in Figure 2.

2) As the main strength of the approach is to localize infiltrating immune cells it would be good to improve on the in vivo images shown in Figure 9 a-c. While the T cell localization is well shown in panel D (which however is obtained by confocal imaging of vibratome sections that does not require an in vivo label), the outline of the glia limitans is not clearly discernable for me in the example images currently provided. This should be feasible as the glial limitans can be identified much clearer in the example images provided in Figure 10.

Answer: We have revised Figure 9 after performing additional EAE experiments using different imaging settings which allowed to significantly improve the 2P-IVM data quality. Instead of following in vitro activated far-red labelled autoaggressive CD4 T cells we adapted the experimental setting to that employed for CD8 T cell trafficking. To this end we transferred naïve GFP+ CD4 T cells a day prior to inducing EAE allowing for their in vivo activation and expansion and followed their trafficking at the indicated timepoints of the clinical disease. This improved experimental setup allowed for imaging sufficient numbers of GFP+ CD4 T cells and thus for their quantification in the respective CNS compartments. These data have been added to the revised Figure 9. We have furthermore included novel confocal images from brain cryosections collected from these animals post-2P-IVM which are displayed in Figures 9i.

3) In Figure 3, the authors make the point that the end feet of the astrocytes with the AQP4-mRuby are unaffected compared to WT astrocytes. They use a swelling score and they compare using the mean and standard deviation between KI and WT mice. Since it's a scale with qualitative values, it would be better to show proportion of end feet of each type per mouse per genotype, as the numbers are just coding qualitative characteristics.

Answer: We have now added the distribution of the swelling score of the perivascular astrocyte end-feet for the respective genotypes for each individual mouse (Figure 3c). This allows to exclude effects of sample preparation artifacts that might hinder our interpretation. The distribution of scores is comparable between heterozygous Aqp4-mRuby3 and WT mice. In addition, we can exclude the occurrence of potential perfusion effects on WT mice, since none show a bias in all cerebrum regions towards enlarged endfeet, as can be seen in mouse 4 that only shows a high number of enlarged endfeet in the neocortex, compared to cerebellum and brainstem.

Reviewer #4 (Remarks to the Author):

Answer: We thank this Reviewer for contributing to improving our manuscript.

Reviewer #5 (Remarks to the Author):

Answer: We thank this Reviewer for contributing to improving our manuscript.

We hope to have answered all queries of the Reviewers in a satisfactory manner and that our revised study is now suitable for publication in *Nature Communications*.

We look forward to your reply.

Sincerely

Dr. rer. physiol. Britta Engelhardt
Professor for Immunobiology

KNOWLEDGE
CREATES
VALUE.

We thank the Reviewers for the positive feedback and we have answered the remaining comments from Reviewer #2 as explained in our point-by-point reply below. Changes in our manuscript have been highlighted in blue.

Reviewer #2:

The resubmission of 'In vivo imaging of the barrier properties of glia limitans during health and neuroinflammation' by Hélie-Legoupil, Kloster et al, I believe, has improved upon their overall characterization of the Aqp4-mRuby3 knock-in mice to study barrier compartments within the CNS. Especially regarding the detailed analysis of T cells within the various barrier compartments and their dynamics during immune surveillance and neuroinflammation. The addition of serial block face electron microscopy to show the composition of the superficial glia limitans does highlight the complexity of astrocytes creating that tissue as well. I do have a few lingering thoughts on our discussion.

1. The authors adequately addressed the possibility for differences between astrocytic endfeet swelling among individual WT and het mice for the perivascular glia limitans and demonstrated that potentially poor tissue fixation in WTs was unlikely to mask a swelling effect in the het mice. Further, the authors nicely add volumetric electron microscopy (EM) and show qualitatively that astrocytes and their processes are likely unaffected in the het mice along the superficial glia limitans. I realize these were performed on one animal per genotype given the constraints with volume EM, as well as the necessity of this approach due to the complexity of the glia limitans there. Is it possible to provide some sort of quantification to accompany the conclusions made on the volume EM here? Perhaps measuring the volume of individual cell bodies and processes among the two animals?

Answer: We thank the Reviewer for the positive feed-back and the additional considerations. Measuring cell body volume is in principle possible but with the material at hand we are not sure if the glial cells are fully contained in the volume. If this is (most probably) not the case - as this was not an aim in the original study design- we obviously cannot get meaningful volume values. In addition, with the material at hand we would only be able to quantify a couple of cells. Thus addressing this question appropriately would require novel tissue preparations and segmentations and thus a significant amount of experimental and analysis work in addition to sacrificing additional mice. We do hope that this Reviewer agrees with us that such an in depth analysis is beyond the scope of the present manuscript and should rather be addressed in a follow-up study.

2. The addition of laminin to the images in Fig. 2g helps greatly to demonstrate the perivascular cuffing occurring in EAE. Do you see similar amounts of perivascular cuffing to this magnitude in WT vs. Aqp4-mRuby het mice with EAE? This would bolster the conclusion that heterozygous Aqp4-mRuby mice have comparable neuroinflammatory responses.

Answer: We thank the Reviewer for this additional suggestion and have quantified the inflammatory cuffs in the brain of Aqp4-mRuby3 and control mice at the chronic stage of EAE. Please note in order to consider the 3R rules we have not set up a novel EAE experiment which in Switzerland is considered severity grade 3 – and thus counted the inflammatory cuffs in tissue specimen still available to us. The novel Figure 2h shows the summary of the quantification of the perivascular cuffs in brain cryosections of WT (n=3) and Aqp4-mRuby3 mice (n=2) at chronic stage of aEAE.

3. For the possible complications of anesthesia altering CSF flow, I appreciate the discussion regarding the low-dose isoflurane potentially having minimal effect on CSF flow. I missed the part where the animals were also under fentanyl/midazolam/medetomidine mixture as well. We use the exact same dosage in our animals for surgeries as well and it very much slows down breathing and heart rate, both of which are known drivers of CSF mobility throughout the brain. Unless the authors have heart rate and breathing

metrics to accompany these experiments indicating these physiological processes were functioning similarly to an awake mouse during the 2P imaging, I think it would be best if the authors omit their conclusions in challenging the entry of CSF into the periarterial spaces here and add the details of the F/M/M cocktail on heart and breathing physiology. There are too many parameters that are altered in these animals in these experiments to conclude that the 10kDa FITC and/or beads would not have ever entered the periarterial spaces and thus challenging the entry of CSF into the periarterial spaces.

Answer: We thank the Reviewer for the additional thoughts on study design and data interpretation of CSF tracer movement as observed by in vivo imaging and we agree that we should be very careful on data interpretation and conclusions and balance our observations with those made by others. We all face the issue that in vivo imaging studies on tracer distribution in the CSF use experimental protocols that will affect physiological parameters. Even imaging tracer distribution in the CSF in awake mice making use of head-fixed mice will cause anxiety.

As suggested by the Reviewer we have added detailed information on the heart rate of the mice as measured throughout the experiments from surgery to imaging in Methods. Please note that our animals were under mechanical ventilation with a gas mix of air and oxygen-containing 0.5–1% isoflurane the breathing rate is fixed at a tidal volume of 180 ml and 110 strokes/min. We have also better highlighted that we do not claim to image under physiological conditions. Indeed we observed reduced heart rates (149 to 196 bpm) during the time period that well correlates with the half lives of the F/M/M cocktail with heart rates increasing to 320 to 400 bpm during imaging, which are however still lower than in awake mice (see figure below). Based on the study by Hablitz et al., a heart rate in this range allows for observation of intermediate “glymphatic” flow. Under the conditions used in our study we did not see any tracer distribution that can be characterized as “glymphatic” flow.

Figure for Reviewer: Temperature and heart rate protocols of 3 mice included in the tracer experiment

74862				83942				83948			
Intervention	Time	Temperature	HR (bpm)	Intervention	Time	Temperature	HR (bpm)	Intervention	Time	Temperature	HR (bpm)
MMF injection	09:10	N/A	N/A	MMF injectio	13:55	N/A	N/A	MMF injectid	09:15	N/A	N/A
Start of Isoflurane	09:40	N/A	N/A	Start of Isoflu	14:20	N/A	N/A	Start of Isoflu	09:35	N/A	N/A
Start ECG record	09:46	32.7°	154	Start ECG rec	14:25	33.4	149	Start ECG rec	09:40	32.2°	196
End of Surgery	10:20	36.4°	444	End of Surger	15:05	36.5	343	End of Surger	10:20	36.4°	346
CM injection	11:25	36.2°	393	CM injection	15:38	36.4	375	CM injection	11:15	36.5°	323
Imaging	11:50	36.5°	400	Imaging	16:00	36.3	368	Imaging	11:45	36.3°	320
Imaging	12:10	36.6°	450	Imaging	16:30	34.6	373	Imaging	12:15	36.2°	336
K/X overdose	12:20	36.4°	460	K/X overdose	17:10	36.1	375	K/X overdose	12:28	36.3°	251

	Half life (im)
Fentanyl	2 to 4h
Medetomidin	30 to 90 min
Midilazolam	30 to min

This Reviewer points out that the FMM cocktail will reduce heart and breathing rates and thus affect “known drivers of CSF mobility throughout the brain”. In apparent contrast the study of Hablitz et al., Sci. Adv. 2019; 5: eaav5447 from 27 February 2019 showed the opposite, namely that “glymphatic” flow increases under conditions of anesthesia and showed that the most “glymphatic” flow was observed under anesthesia producing the lowest heart rate. In that study, mice treated with ketamine/xylazine had the lowest heart rate (just below 300bpm) and showed the greatest tracer influx (Figure 4 in their study). Thus we think that additional studies will be needed to clarify these apparent different observations. Please also note that the volumes injected in the different studies also significantly vary. In the study of Hablitz et al a total of 10 ml of volume was infused into the cisterna magna within 5 minutes while our protocol only infuses a total of 2.5 ml over 2.5 minutes. Thus application of the lower volume at a lower

infusion rate as used by us will cause additional apparent differences to observations previously made with higher infusion rates and volumes. While we think this is an exciting and important topic, solving these issues is beyond the scope of this manuscript but we hope that the Aqp4-mRuby3 mouse will find interest in the community and help to further explore CSF flow in the brain. We have adapted our paragraph in the discussion accordingly.

We hope to have answered all queries of the Reviewer in a satisfactory manner and that our revised study is now suitable for publication in *Nature Communications*